# A gain of function mutation in *SlNRC4a* enhances basal immunity resulting in broad-spectrum disease resistance

Lorena Pizarro[1,2,6], Meirav Leibman-Markus[1,6], Rupali Gupta[1], Neta Kovetz[1], Ilana Shtein [3], Einat Bar[4], Rachel Davidovich-Rikanati[4], Raz Zarivach[5], Efraim Lewinsohn[4], Adi Avni[2] & Maya Bar [1✉]

Plants rely on innate immunity to perceive and ward off microbes and pests, and are able to overcome the majority of invading microorganisms. Even so, specialized pathogens overcome plant defenses, posing a persistent threat to crop and food security worldwide, raising the need for agricultural products with broad, efficient resistance. Here we report a specific mutation in a tomato (*S. lycopersicum*) helper nucleotide-binding domain leucine-rich repeat H-NLR, *SlNRC4a*, which results in gain of function constitutive basal defense activation, in absence of PRR activation. Knockout of the entire NRC4 clade in tomato was reported to compromise Rpi-blb2 mediated immunity. The SlNRC4a mutant reported here possesses enhanced immunity and disease resistance to a broad-spectrum of pathogenic fungi, bacteria and pests, while lacking auto-activated HR or negative effects on plant growth and crop yield, providing promising prospects for agricultural adaptation in the war against plant pathogens that decrease productivity.

[1] Department of Plant Pathology and Weed Research, ARO, Volcani Center, Rishon LeZion 7505101, Israel. [2] School of Plant Sciences and Food Security, Tel Aviv University, Tel Aviv 69978, Israel. [3] Department of Oenology and Agriculture, Eastern Region R&D Center, Ariel, Israel. [4] Department of Vegetable Crops, Newe Ya'ar Research Center, The Volcani Center, Ramat Yishay 30095, Israel. [5] Department of Life Sciences, Faculty of Natural Sciences, The National Institute for Biotechnology in the Negev, and Ilse Katz Institute for Nanoscale Science & Technology, Ben-Gurion University of the Negev, Beer Sheva 8410501, Israel. [6] These authors contributed equally: Lorena Pizarro, Meirav Leibman-Markus. ✉email: mayabar@volcani.agri.gov.il

Plants are constantly challenged by a plethora of potential pathogens—fungi, oomycetes, bacteria, viruses, nematodes, and insects. To defend themselves, plants utilize an innate two-tiered immune system: pattern triggered immunity (PTI) and effector-triggered immunity (ETI)[1]. The first line is formed by pattern recognition receptors (PRRs), located at the cell surface, that recognize microbe-associated molecular patterns (MAMPs), leading to PTI[2]. Upon MAMP recognition, PRRs activate a signaling cascade, that leads to robust transcriptional changes and physiological changes in order to restrict pathogen attack[3]. This first defense layer is able to control most non-adapted pathogens. During infection of adapted pathogens, PTI contributes to basal immunity[4]. Adapted pathogens have evolved mechanisms to repress PTI by delivering proteins (effectors), surpassing the first tier of immunity[5,6]. Resistant plants have developed a second tier of immunity, responsible for warding off adapted pathogens in a strain-specific interaction[7,8]. This resistance is based on resistance (R) proteins that recognize specific effector (Avr) proteins, leading to ETI[9]. ETI activation leads to signaling cascades which restore plant resistance[7,8].

R proteins can be classified, based on their structure, to kinase proteins and nucleotide binding (NB) leucine-rich repeat proteins (NLR)[10]. NLRs typically consist of a highly polymorphic C-terminal LRR domain, that is thought to confer recognition specificity, and a central NB-ARC domain, that is thought to function as a molecular switch[10]. NLRs can be further classified into two subgroups, based on their N-terminal domain: NLRs containing a toll-interleukin 1 receptor (TIR) domain (TNLs), or a coiled coil (CC) domain (CNLs)[10]. One arising model suggests NLRs function in pairs, with one acting as a "sensor" (s-NLR), enabling effector recognition, and the other as a "helper" (H-NLR), mediating signal transduction, forming an interconnected immune-network[11,12]. Several regulatory mechanisms of NLR activity have been demonstrated, including intramolecular regulation and homo- and heterodimerization[11,12]. Recently, the structure of ZAR1, a CNL-type NLR, has been elucidated, demonstrating a formation of a pentameric α-helical barrel complex that is reshaped, after effector interaction, leading to exposure of N-terminal amphipathic α-helices that form a plasma membrane pore that is proposed to enable the initiation of programmed cell death (PCD)[13,14].

In Solanaceae, a subfamily of NLRs termed NLR required for cell death (NRC) emerges as a key family of H-NLRs, required for downstream signaling of multiple s-NLRs[12]. Interestingly, H-NLRs can mediate MAMP and effector signaling sensed by several LRR-receptors[15–17]. In tomato, the PRR LeEIX2 recognizes the fungal MAMP EIX, and triggers defense responses[18]. Recently, we reported that the tomato NLR—NRC4a functions as an H-NLR and is required for LeEIX2-mediated signaling[19]. Overexpression of NRC4a or of its CC domain (CCd), enhances EIX and flg22 defense responses, while virus induced gene silencing of the SlNRC4 clade (targeting SlNRC4a, SlNRC4b, and SlNRC4c) diminishes EIX-mediated defense responses. Most interestingly, NRC4a CRISPRed plants, encoding a 67 aa truncated protein variant, resulted in a gain of function mutant that displayed intensified defense responses when challenged with EIX[19]. Interestingly, recent work described a near-full NRC4 clade deletion in tomato, in which immunity mediated by the *Solanum bulbocastanum* late blight resistance protein Rpi-blb2 was abolished, but flg22 mediated responses were retained[20].

SlNRC4a emerges as a H-NLR, integrating multiple signals sensed by both PRRs and s-NLRs[11,17,21]. In this work, we characterize the gain of function SlNRC4a mutant, demonstrating that it possesses broad-spectrum disease resistance in tomato, and unravel some of the cellular and physiological processes enabling this broad resistance. We conclude that in our mutant, the remaining 67 amino acids of NRC4a provide gain of function immuno-activation and increased defense resistance, without generating auto-active HR as was reported for some NLR gain of function mutants[22], allowing for interesting prospects in future agricultural applications.

## Results

### slnrc4a mutant possesses broad-spectrum pathogen resistance.
We generated two independent *slnrc4a* edited lines, *slnrc4a-2* and *slnrc4a-5*, with an insertion of a cytosine and a thymine, respectively, resulting in 67 aa truncated variants[19]. Both lines display similar behavior, acting as gain of function mutants, and are used jointly in all experiments throughout this work. The mutant lines display a stronger response to the elicitors flg22 and EIX, acting as a gain of function of SlNRC4a. We hypothesized that *slnrc4a* gain of function will lead to enhanced basal immunity, resulting in broad-spectrum pathogen and pest resistance. Diverse tomato pathogens were tested to determine the *slnrc4a* resistance-spectrum (Fig. 1). For bacteria, we examined *Pseudomonas syringae pv. tomato DC3000* (*Pst*) and *Xanthomonas campestris pv. vesicatora* (*Xcv*). *Xcv* causes bacterial leaf spot (BLS) on peppers and tomatoes. BLS severely reduces plant productivity and yield[23]. *Pst* causes bacterial speck (BL), reducing yield and heavily damaging fruit quality and marketability[24]. Tomato wild type (WT) and *slnrc4a* plants challenged with *Xcv* and *Pst* demonstrated enhanced resistance against *Xcv* and *Pst* in the *slnrc4a* gain of function mutant, leading to 71% and 46% decrease in bacterial growth compared with WT, respectively (Fig. 1a–c). We analyzed the capacity of necrotrophic fungal pathogens to infect the *slnrc4a* edited tomato line, utilizing *Botrytis cinerea* (*Bc*) and *Sclerotinia sclerotiorum* (*Ss*) that cause gray and white mold diseases, respectively[25,26]. Both fungi have a wide range of hosts, many of them crop plants, and cause severe damage in greenhouses and post-harvest storage and transport[25,26]. Both fungi exhibited reduced symptoms on *slnrc4a*, leading to a decrease of 35% and 85% in lesion area, respectively (Fig. 1d–f, Supplementary Fig. 1a, b). In order to examine the ability of the mutant plants to resist a biotrophic fungus, we utilized the powdery mildew fungus *Oidium neolycopersici* (*On*). Powdery mildew infection destroys vegetative plant tissue and leads to the production of non-marketable fruit[27]. We observed a 65% decrease in *On* infected leaf area in *slnrc4a* (Fig. 1j, k, Supplementary Fig. 1c). We examined pest resistance by exposing plants to *Bemisia tabaci* (whitefly) and *Tuta absoluta* (tomato leafminer). Both insects cause significant damage in tomato cultivation; *B. tabaci* is responsible for transmission of viral disease[28,29]. As in the case of microbial attack, *slnrc4a* plants were better at warding off pests, with a 73% reduction in whitefly numbers per leaflet (Fig. 1g) and 87% reduction of *T. absoluta*- infected leaf area (Fig. 1h, i, Supplementary Fig. 1d). Conforming to our hypothesis, the *slnrc4a* gain of function mutant showed a remarkable increase in resistance across a wide range of biotic stressors with different life-styles and attack strategies including bacteria, necrotrophic fungi, a biotrophic fungus, and pests.

### slnrc4a possesses increased basal defense.
One of the mechanisms underlying the broad-spectrum resistance we observed in the *slnrc4a* gain of function mutant could be a constitutive increase in basal immunity. In order to verify this hypothesis, we tested defense parameters in steady state. Upon microbial recognition, a rapid accumulation of extracellular reactive oxygen species (ROS), an oxidative burst, occurs[30], likely triggering downstream defense responses: defense gene expression, hypersensitive response (HR), callose deposition and systemic acquired

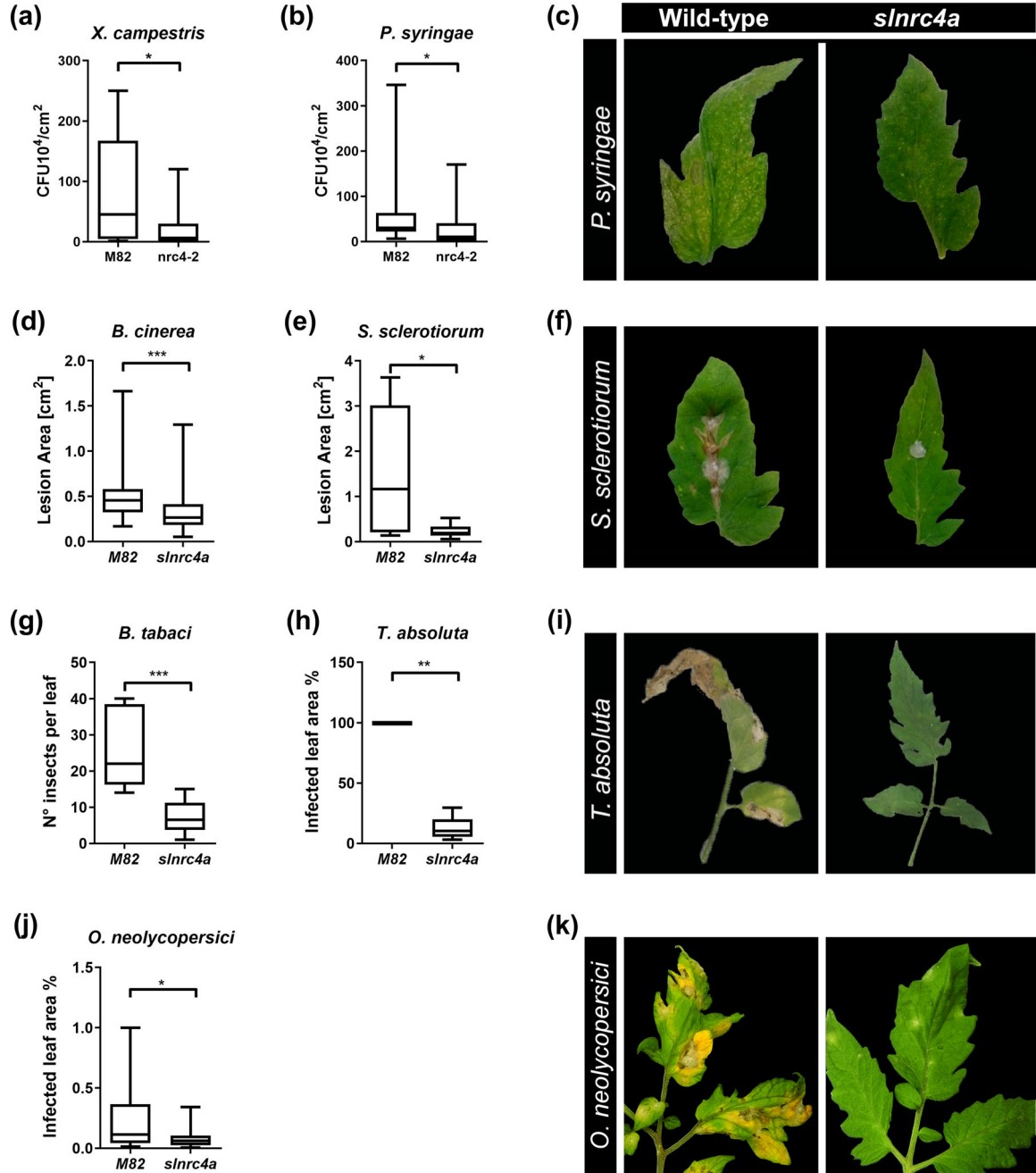

**Fig. 1 *slnrc4a* gain of function mutant presents enhanced resistance to diverse plant pathogens and pests.** WT (cv M82) and *slnrc4a* plants were challenged with several pathogens and pests. Bacterial growth (CFU) was measured 3 days after inoculation with (**a**) *X. campestris* ($10X^5$ CFU/mL) and (**b**) *P. syringae* ($10X^5$ CFU/mL). **c** Representative images of *P. syringae* symptomatic leaves. The lesion area was measured 3 days after inoculation with (**d**) *B. cinerea* ($10X^6$ spores/mL) (**e**) *S. sclerotium* ($10X^6$ spores/mL). **f** Representative images of *S. sclerotium* symptomatic leaves. Infestation was determined by counting number of insects per leaf and measuring % of infected leaf area two-weeks after (**g**) *B. tabaci* and (**h**) *T. absoluta* exposure, respectively. **i** Representative images of *T. absoluta* symptomatic leaves. **j** *O. neolycopersici* infection was measured as percentage of infected leaf out of total leaf area. **k** Representative images of *O. neolycopersici* symptomatic leaves. Average ± SEM of 3–4 independent replicates is shown. **a**, **b**, **d**, **e**, **g**, **h**, **j** Boxplots are shown with the inter-quartile-ranges (box), medians (black line in box) and outer quartile whiskers, minimum to maximum values. Asterisks represent statistical significance in *t*-test with welch correction (\**p*-value <0.05; \*\**p*-value < 0.01; \*\*\**p*-value < 0.001).

resistance (SAR)[30]. Electrolyte leakage is a stress hallmark, that can evolve into programed cell death (PCD) such as pathogen-associated HR[31]. Ethylene production is triggered in both PTI and ETI[32], with ethylene regulating defense response hormonal cross-talk[33]. Defense responses also include callose deposition, where plants form a physical barrier to prevent pathogen invasion[34].

ROS was measured in WT and *slnrc4a* plants in steady state conditions (Fig. 2). We did not detect an oxidative burst, and no difference in ROS levels were observed in basal states between WT and *slnrc4a* plants (Fig. 2a, b). The *slnrc4a* gain of function mutation does not lead to ligand-independent electrolyte leakage (Fig. 2c, Supplementary Fig. 1f), consistent with it not resulting in auto-activated HR[19]. Our mutant plants are healthy and develop

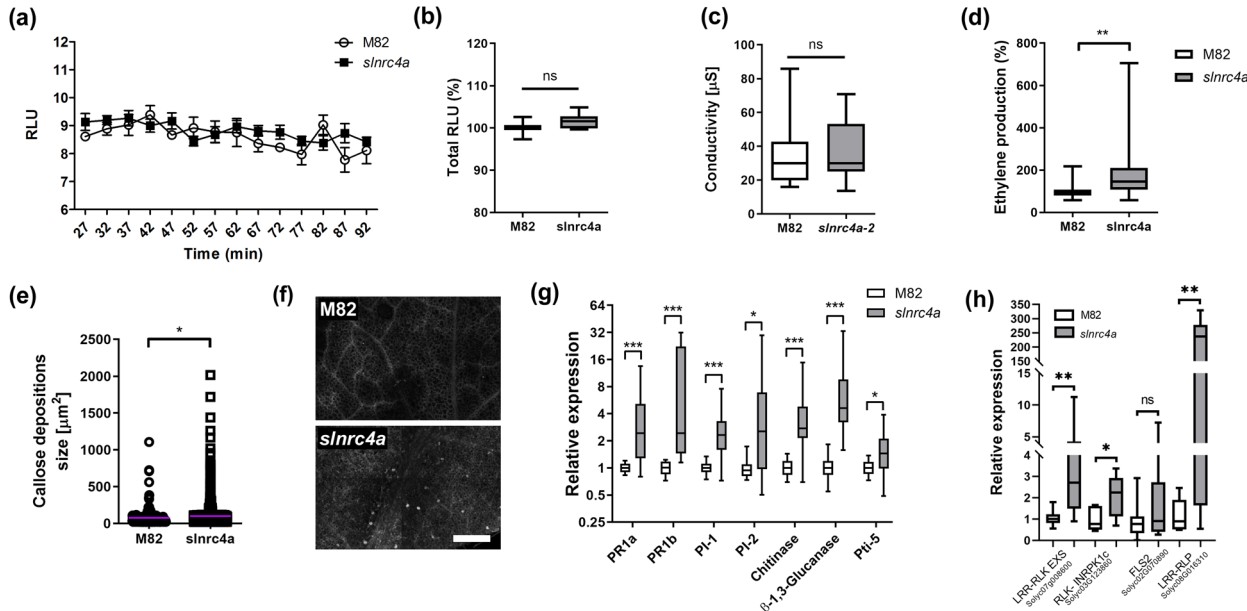

**Fig. 2 slnrc4a gain of function mutant has elevated defense parameters in steady state condition. a** ROS production was measured every 5 min for 90 min in WT (cv M82) and *slnrc4a* line using the HRP-luminol method. Average ± SEM of four independent replicates is shown (two-way ANOVA, no significant difference). **b** Total ROS production presented in **a**. Average ± SEM of four independent replicates is shown (one-way ANOVA, no significant difference). **c** Conductivity levels of M82 and *slnrc4a* samples immersed in water for 24 h was measured. Average ± SEM of four independent replicates is shown (one-way ANOVA, no significant difference). **d** Ethylene production of M82 and *slnrc4a* samples was measured using gas chromatography. M82 average ethylene production is defined as 100%. Average ± SEM of five independent experiments is presented. Asterisks represent statistical significance in *t*-test with welch correction (**$p$-value < 0.01). **e** Callose deposition was observed by confocal microscopy after aniline blue staining of M82 and *slnrc4a* leaf discs. Callose deposit size was measured using the object counting tool of ImageJ. Average ± SEM of three independent replicates is shown, all individual values presented, line indicates mean. Asterisks represent statistical significance in *t*-test with welch correction (*$p$-value <0.05). **f** Callose deposition representative images. Scale bar = 200 μm. **g** Gene expression analysis of pathogen responsive genes in M82 and *slnrc4a* plants was measured by RT-qPCR. Relative expression normalized to M82. Average ± SEM of three independent replicates is shown. Asterisks represent statistical significance in *t*-test with welch correction comparing each gene (*$p$-value < 0.05; ***$p$-value < 0.001). **h** Gene expression analysis of PRR genes in M82 and *slnrc4a* plants was measured by RT-qPCR. Relative expression normalized to M82. Average ± SEM of three independent replicates is shown. Asterisks represent statistical significance in *t*-test with Welch's correction comparing each gene (*$p$-value < 0.05; **$p$-value < 0.01). **b–d**, **g**, **h** Boxplots are shown with the inter-quartile-ranges (box), medians (black line in box) and outer quartile whiskers, minimum to maximum values.

normally. However, we detected a steady state, ligand-independent increase of 46% in ethylene in *slnrc4a* lines when compared to the basal ethylene level of WT plants (Fig. 2d, Supplementary Fig. 1e). Similarly, *slnrc4a* lines also displayed a pre-deposition of callose, with a 30% increase of leaf epidermal callose foci in basal conditions when compared to WT (Fig. 2e, f). In *slnrc4a* lines, we observed an increase in steady state defense parameters such as ethylene biosynthesis and callose deposition, However, no increase was observed in responses that have the potential to cause HR, such as ROS and electrolyte leakage. Indeed, overexpression of SlNRC4a's CCd or *slnrc4a* truncated-mutant peptide, does not promote ligand-independent HR[19], as was reported in certain NLR auto-activation gain of function mutants[22].

The elevated defense parameters detected in basal conditions likely enable the broad resistance exhibited by the *slnrc4a* gain of function lines. To further investigate the basis of this mechanism, we profiled *slnrc4a* pathogen responsive gene expression. Seven genes were analyzed under basal conditions. As pathogenesis-related protein 1 (PR-1) family members are abundantly produced upon pathogen attack, and PR-1 gene expression is commonly used as a defense marker, we tested the relative expression of both *PR-1a* (Solyc01g106620) and *PR-1b* (Solyc00g174340). Additionally, we quantified the mRNA level of *pathogen induced 1* (PI-1, Solyc01g097270), *proteinase inhibitor 2* (PI-2, Solyc03g020080), a *chitinase* (Solyc10g055800), a *β-1, 3- glucanase* (Solyc01g060020), and Pto-interacting 5 (*Pti-*

5, Solyc02g077370), all of which are reported to undergo induction after exposure to pathogens[35–38]. PI-1 and PI-2 are JA responsive and considered markers of induced systemic resistance (ISR)[39–42]. Pti5 is ethylene responsive[43]. PR1a is SA responsive and considered SAR related[39,44]. PR1b is upregulated by both SAR and ISR activation[37,45]. Distinctions between ISR and SAR are not clear cut in tomato, and they can overlap[46,47]. The expression of all seven genes increased by two to nine-fold in *slnrc4a* plants (Fig. 2g, Supplementary Fig. 1g). We compared PRR gene expression in M82 and *slnrc4a*. Four genes were analyzed under basal conditions (Fig. 2h). *LRR-RLK-EXS* (Solyc07g008600), *RLK-INRPK1c* (Solyc03G123860), and an LRR-RLP (Solyc08G016310)- were all significantly upregulated in *slnrc4a*, while the LRR-RLK FLS2 (Solyc02G070890) was not. The induction of PRRs and pathogen responsive genes in *slnrc4a* plants under steady state conditions likely provides the leverage needed for resistance.

To further investigate the mechanisms enabling broad pathogen resistance in the *slnrc4a* gain of function mutant, volatile metabolites were quantified using GC-MS analysis in *slnrc4a* and WT plants (Fig. 3). Increased levels of several volatile compounds including sesquiterpenes, monoterpenes, and phenylpropanoids were found in *slnrc4a* lines compared to WT (Fig. 3a, Supplementary Table 1). Herbivore-induced terpenoid emissions have been shown to induce defense genes and activate jasmonate, ethylene and calcium signaling[48]. For example, *E*-caryophyllene— a sesquiterpene which plays an important role in plant defense[49],

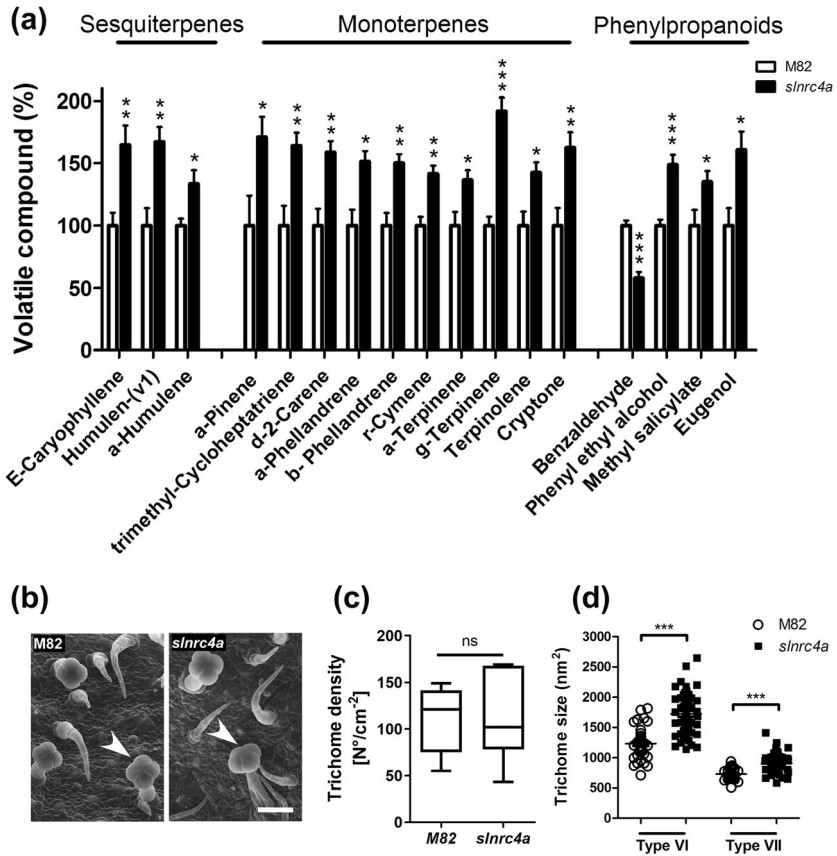

**Fig. 3 slnrc4a gain of function mutant has elevated defense metabolites and increased trichome head size. a** Volatile metabolome was performed for M82 and slnrc4a. Compound level was normalized to M82. Significantly different Sesquiterpenes, monoterpenes and phenylpropanoids are shown. Average ± SEM of three independent replicates is shown. Asterisks represent statistical significance in t-test with welch correction for each compound (*p-value < 0.05; **p-value < 0.01; ***p-value < 0.001). **b** Scanning electron microscope images of young M82 and slnrc4a leaves of Type VI glandular trichomes with four head cells. Arrowheads indicated type VI glandular trichome. Scale bar = 50 μm. **c, d** Trichome density and size was determined using ImageJ. Average ± SEM of three independent replicates is shown. Asterisks represent statistical significance in t-test with welch correction for each trichome type (***p-value < 0.001). **c** Boxplot with inter-quartile-ranges (box), medians (black line in box) and outer quartile whiskers, minimum to maximum values. **d** All individual values presented, line indicates mean.

was elevated by 65% in the slnrc4a gain of function mutant (Fig. 3a, Supplementary Table 1). Monoterpenes have been implicated in promoting plant defense. We found a 71% enhancement in emission of α-pinene in slnrc4a (Fig. 3a, Supplementary Table 1). Interestingly, increased emission of α-pinene was reported in arabidopsis during SAR[50], while exogenous application of α-pinene led to induction of SAR-related genes and resistance to P. syringae[50]. Phenylpropanoids play important roles in resistance to pathogen attack[51]. Interestingly, methyl salicylate (MeSA), a methylated volatile derivative of salicylic acid (SA), is involved in induction of systemic plant resistance[51]. slnrc4a plants displayed a 35% increase in MeSA while having a 42% decrease in the MeSA precursor—benzaldhyde (Fig. 3a, Supplementary Table 1), pointing to a metabolic shift toward the production of MeSA and SA, suggesting steady state activation of SA mediated defense pathways in slnrc4a.

Type VI trichomes represent the most abundant trichome type on leaves and stems of tomato plants, and significantly contribute to plant defense. The volatile compounds and proteinases produced in these trichomes have been reported in many studies to be toxic to insects, fungi, and bacteria[52]. Type VI glandular trichomes produce large amounts of terpenes[53,54]. Considering the higher level of sesquiterpenes and monoterpenes detected in slnrc4a, we analyzed trichome morphology and distribution

(Fig. 3b–d). slnrc4a and WT plants exhibited similar trichome density, yet trichome types VI and VII possessed larger heads in slnrc4a plants by 40% and 20%, respectively (Fig. 3b–d). This increase in trichome head size likely results in the measured increased emissions.

**slnrc4a enhances defense responses upon elicitation.** In steady state conditions, slnrc4a plants produce elevated levels of ethylene and callose, higher mRNA levels of pathogen responsive genes, and increased volatile compound/secondary metabolite emission (Figs. 2 and 3). We propose that these changes underlie a constant immuno-activated state that enables the broad-spectrum pathogen resistance of slnrc4a plants. We anticipated that this would likely also facilitate mounting of defense responses upon elicitation. We tested defense responses upon elicitation with the fungal MAMP-EIX (Supplementary Fig. 2).

ROS measurements immediately after elicitation with EIX displayed enhancement of 200% in oxidative burst in slnrc4a, compared to that of the elicited WT, in line with our previous report (Supplementary Fig. 2a)[19]. Ethylene production and callose deposition were dramatically increased in response to EIX in slnrc4a plants compared to control (Supplementary Fig. 2b, d, e). We observed a 200% increase in ethylene production, in agreement with our previous report[19], and higher accumulation of callose, leading to a 60% increase in depositions

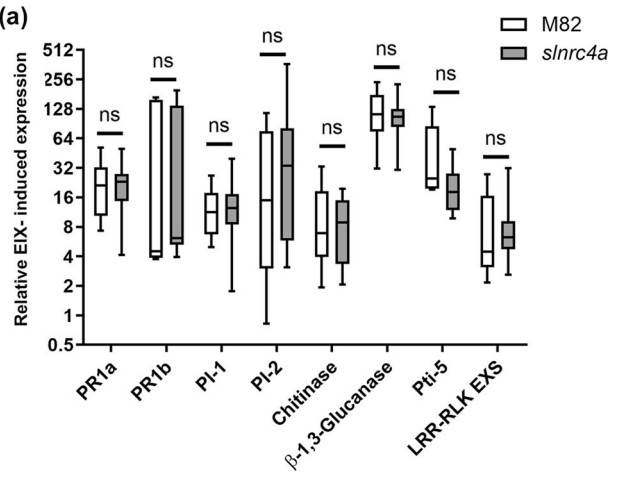

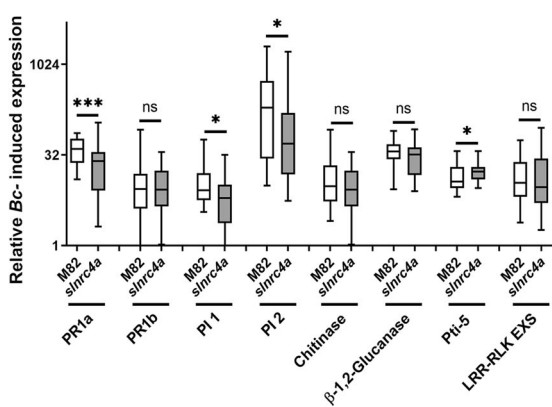

**Fig. 4 Defense gene induction is similar in M82 and *slnrc4a* in response to MAMP elicitation and pathogen inoculation. a** Gene expression analysis of pathogen responsive genes was measured by RT-qPCR in M82 and *slnrc4a* plants 24 h after petiole application of EIX. Relative expression normalized to control M82 (mock). Average ± SEM of three independent experiments is shown. No statistically significant differences were observed among WT and *slnrc4a* (*t*-test, welch correction). **b** Gene expression analysis of pathogen responsive genes was measured by RT-qPCR in M82 and *slnrc4a* samples 24 h after *Bc* inoculation. Relative expression normalized to M82 (mock). Average ± SEM of three independent experiments is shown. Asterisks represent statistical significance in *t*-test with welch correction comparing each gene (*$p$-value < 0.05; ***$p$-value < 0.001). Boxplots are shown with inter-quartile-ranges (box), medians (black line in box) and outer quartile whiskers, minimum to maximum values.

size, in *slnrc4a* compared to WT (Supplementary Fig. 2b, d, e). No significant conductivity increase was detected in *slnrc4a* plants 24 h after elicitation when compared to WT (Supplementary Fig. 2c). The overall enhancement of defense responses observed upon MAMP elicitation can be explained by the increased basal immunity in steady state (Fig. 2), correlating with *slnrc4a* plants increased resistance to diverse pathogens (Fig. 1).

***slnrc4a* elicitor and pathogen-induced gene expression.** Enhanced steady state defense parameters in *slnrc4a* correlated with increased disease resistance (Fig. 1) and increased EIX response (Supplementary Fig. 2). To investigate genetic mechanisms of elicitation and disease in *slnrc4a*, we profiled pathogen responsive gene expression (Fig. 4), assaying the eight genes tested in steady state conditions (Fig. 2), in response to EIX elicitation (Fig. 4a) and *B. cinerea* inoculation (Fig. 4b). The

tested genes were highly induced upon elicitation or *B. cinerea* inoculation, with no significant differences between WT and *slnrc4a* plants in EIX response (Fig. 4a), and some small but significant differences between WT and *slnrc4a* plants in *B. cinerea* response (Fig. 4b). We observed a two to nine-fold increase in defense gene mRNA levels in *slnrc4a* plants in steady state conditions (Fig. 2g); an induction of up to 122–fold in mRNA levels after EIX treatment, in both WT and *slnrc4a* plants (Fig.4a); and an induction of up to ~500 fold in mRNA levels after *B. cinerea* inoculation (Fig. 4b). Correlating with lower *B. cinerea* disease levels, we observed a small but significant decrease in the induction of some of the assayed genes following *B. cinerea* inoculation, in *slnrc4a* plants when compared with WT plants (PR-1a, PI-1, PI-2, Fig. 4b). Induction of the ethylene pathway gene Pti5 was slightly increased in *slnrc4a* plants when compared with WT plants in response to *B. cinerea* inoculation. The expression levels of some of these defense genes, such as chitinase and PI-2, has been previously correlated with disease levels during *B. cinerea* infection[37,55]: plants with more severe symptoms have higher expression of these genes.

**No significant yield cost to *slnrc4a* enhanced basal immunity.** Crop diseases are a major threat to global food security. As plant growth and immunity pathways are intertwined, resistance may be correlated with reduction in growth and yield[56]. The need for both resistant and high yield agricultural plants led us to examine whether the broad-spectrum resistance of *slnrc4a* came with an unwanted price. Several yield parameters were measured and compared between WT and *slnrc4a* plants (Fig. 5). No differences were observed in total fruit number (Fig. 5a, Supplementary Fig. 1h) or total fruit weight (Fig. 5b, Supplementary Fig. 1i), ripe fruit number (Fig. 5c, Supplementary Fig. 1j) or ripe fruit weight (Fig. 5d, Supplementary Fig. 1k), plant fresh weight (Fig. 5e, Supplementary Fig. 1l), Harvest Index (Fig. 5f, Supplementary Fig. 1m) or soluble sugar content in mature tomato fruits (Fig. 5g, Supplementary Fig. 1n), between WT and *slnrc4a* plants (Fig. 5). We conclude that the broad resistance conferred by the *slnrc4a* gain of function mutation does not cause a measurable yield cost under lab and greenhouse conditions.

**Cas9-free *slnrc4a* line maintains disease resistance.** Since *slnrc4a* plants display a promising agricultural combination of high tolerance to a wide variety of pathogens and pests (Fig. 1) with no detectable yield cost (Fig. 5), we wanted to verify that non-transgenic *slnrc4a* progeny retained these desirable properties. Seedlings of the T3 generation were genotyped. *slnrc4a#5-11* lost the cas9 gene, was negative for the selection gene NPTII gene, and possessed the mutation in SlNRC4a (Supplementary Fig. 3a, b). The identified transgene-free *slnrc4a* genotype was verified for increased steady state and elicited defense parameters, and pathogen resistance. As expected, *slnrc4a#5-11* exhibited a robust increase in ethylene production compared to WT, showing 93% increase in steady state conditions (Supplementary Fig. 3c). Upon elicitation, *slnrc4a#5-11* line exhibited a substantial 54% increase in ethylene induction compared to WT plants treated with EIX (Supplementary Fig. 3c). *slnrc4a#5-11* line displayed enhanced resistance to *B. cinerea*, leading to 20% reduction in lesion size (Supplementary Fig. 3d, e). This transgene-free *slnrc4a* line confirms that observed phenotypes in *slnrc4a* lines result from the generated mutation and not from pleiotropic effects stemming from transgene-insertion, and exemplifies the potential of utilizing CRISPR Cas-9 gene editing for sustainable agriculture.

**NRC4 clade deletion vs. the gain of function *slnrc4a* mutant.** A recent report described an NRC4 clade deletion in tomato,

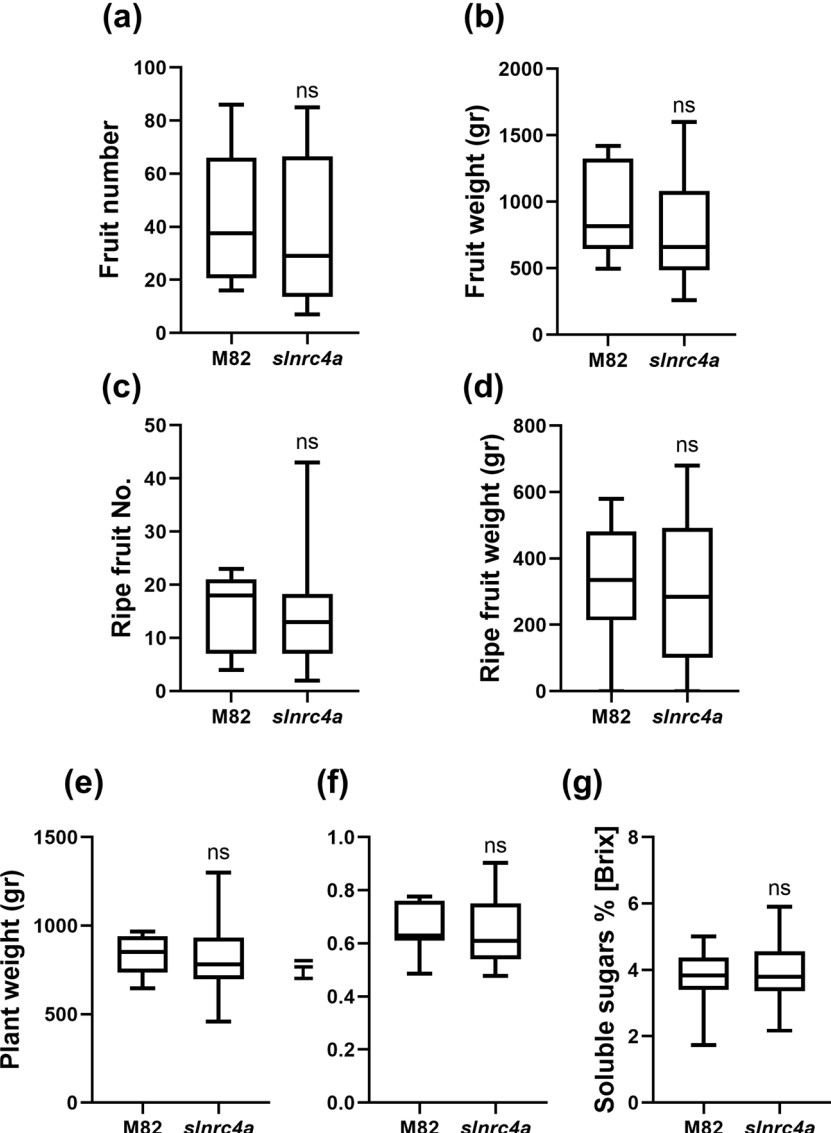

**Fig. 5 *slnrc4a* gain of function mutant line has similar yield production to WT.** Phenotypic parameters of yield were measured in M82 and *slnrc4a* plants. **a** Total fruit number is presented as number of tomato fruits produced by plant. **b** Total fruit weight per plant was measured. **c, d** Same as **a, b**, but ripe fruit only. **e** Total plant fresh weight (aerial tissues only). **f** Harvest index (HI) of plants was calculated as the ratio between the total fruit yield mass and total biomass. **g** Total soluble sugars were measured by refractometry expressed as °Brix. Average ± SEM of three independent replicates is shown. No statistically significant differences were observed among WT and *slnrc4a* (*t*-test, welch correction). Boxplots are shown with inter-quartile-ranges (box), medians (black line in box) and outer quartile whiskers, minimum to maximum values.

wherein Rpi-blb2 mediated immune responses were compromised, but flg22 induced immune responses were not significantly altered[20]. In the mutant from that work, a genomic deletion of 53 Kb results in a full deletion of SlNRC4a, SlNRC4b, and SlNRC5, and a partial deletion of SlNRC4c. We previously reported that silencing members of the NRC4 clade caused a significant decrease in EIX mediated defense responses[19]. In that work, SlNRC4a and SlNRC4b were expressed at ~25% of their levels in WT, and SlNRC4c was expressed at ~35% of its levels in WT. To further investigate the differences between the NRC4 clade deletion mutant and our gain of function NRC4a mutant, we compared EIX induced defense responses and *B. cinerea* disease levels between these lines and their respective backgrounds. We found that the NRC4 near-full clade deletion has a significant reduction in EIX mediated immunity (Fig. 6), as measured by ROS burst (Fig. 6a) and ethylene production (Fig. 6b), consistent with the report for Rpi-blb2, and a significant increase in *B.*

*cinerea* susceptibility (Fig. 6c). Our gain of function *slnrc4a* mutant displayed an increase in EIX mediated immunity (Fig. 6a, b, Supplementary Fig. 2) and decrease in *B. cinerea* susceptibility (Figs. 1d and 6c). To further confirm the responsiveness of the NRC4 clade to EIX elicitation and *B. cinerea* disease, we assayed the clade gene expression levels in response to EIX treatment and *Bc* inoculation, finding that NRC clade members are induced in response to EIX (Supplementary Fig. 4a) and *B. cinerea* (Supplementary Fig. 4b).

**The NRC4 peptide does not require PRR activation.** We previously identified SlNRC4 as an interacting partner for PRRs, with the interaction occurring in the TSM (triton x-100 soluble membrane) fraction. We also demonstrated that the gain of function *slnrc4a* mutant possesses similar levels of *SlNRC4a* mRNA as the WT M82[19]. As we demonstrated here that the 67

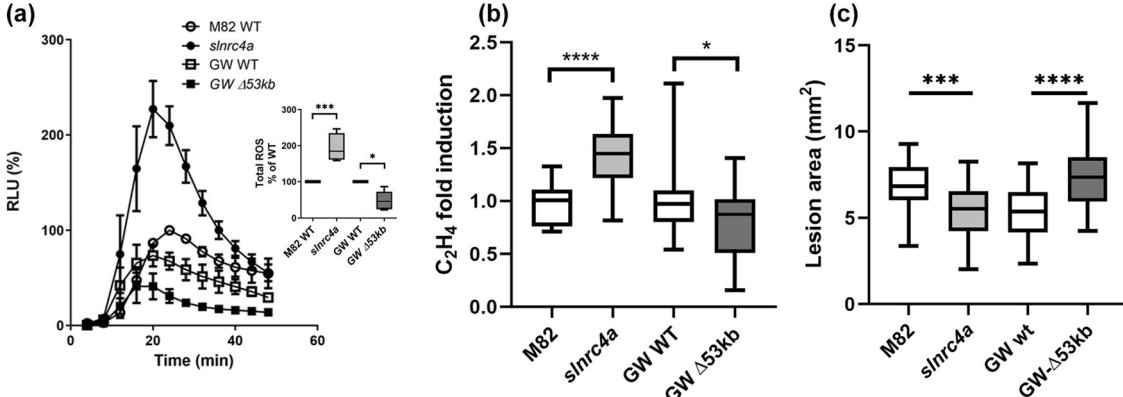

**Fig. 6 Comparison of the NRC4 clade 53Kb deletion and the gain of function *slnrc4a* mutant.** Defense responses elicited by the MAMP EIX and disease susceptibility to *B. cinerea* were compared between the gain of function *slnrc4a* mutant and the recently reported 53 Kb near-full NRC4 clade genomic deletion. **a** ROS production was measured every 5 min for 50 min in WT M82, *slnrc4a*, WT GW, and GWΔ53Kb lines using the HRP-luminol method. Average ± SEM of five independent replicates is shown, N > 48 per genotype. Results were analyzed for statistical significance in a two-way ANOVA, ($p <$ 0.0001). Inset: total ROS production presented in **a**. Average ± SEM of five independent replicates is shown, asterisks represent statistically significant differences in a one-way ANOVA ($p <$ 0.0001) with a Bonferroni post hoc test (*$p$-value < 0.05; ***$p$-value < 0.001). **b** Ethylene production of WT M82, *slnrc4a*, WT GW and GWΔ53Kb samples was measured using gas chromatography. M82 or GW-WT average ethylene production is defined as 1. Average ± SEM of three independent experiments is presented, N > 10 per genotype. Asterisks represent statistical significance in one-way ANOVA ($p <$ 0.0001) with a Dunnett post hoc test (****$p <$ 0.0001; *$p <$ 0.05). **c** Lesion area was measured seven days after inoculation with *B. cinerea* ($10X^6$ spores/mL). Average ± SEM of three experiments (N > 24 per genotype). Asterisks represent statistical significant in a one-way ANOVA with a Tukey post-hoc test, $p <$ 0.0001. ***$p <$ 0.001, ****$p <$ 0.0001. Boxplots are shown with inter-quartile-ranges (box), medians (black line in box) and outer quartile whiskers, minimum to maximum values.

amino acid CRISPR generated peptide induces a constitutively immuno-activated state, but does not auto-activate HR/ cell death, we proceeded to examine whether this peptide retains the ability to associate with the membrane, and whether it requires PRR activation to promote immunity signaling. Recent work has posited that NLRs homologous to NRC can form a "resistosome" structure[13]. Figure 7 presents a model of the SlNRC4 peptide, SlNRC4[1–67], showing that, from a modeling perspective, it retains the ability to achieve the N-terminal NRC4 conformation (Fig. 7a) and can theoretically be associated with the membrane (Fig. 7b). We confirmed its ability to associate with the membrane in fractionation western-blotting assays (Fig. 7c, Supplementary Fig. 6). In agreement, we demonstrate that the 67 amino acid peptide does not require PRR activation to achieve immune signaling: sterile grown *slnrc4a* plants that were not pre-exposed to microorganisms, MAMPs/PAMPs, or biotic stress, retained their increased disease resistance when treated with a pathogen (Fig. 7d), increased defense responses (Fig. 7e), and increased defense gene expression (Fig. 7f) when compared to their identically grown wild-type counterparts, demonstrating that enhanced basal immunity in *slnrc4a* plants is an intrinsic property not dependent on external stimuli.

Our data indicates that while the SlNRC4a 67 amino acid peptide can exhibit MAMP dependency in some cases (Fig. 6, Supplementary Figs. 2 and 3), it does not require MAMPs in order to promote an increase in certain aspects of basal immunity or disease resistance (Figs. 2 and 7). MAMPs mediate signaling via their PRRs. We previously showed that SlNRC4a overexpression can promote an increase in LeEIX2 presence on endosomes[19], where LeEIX2 signaling is taking place[57]. Overexpression of the full length SlNRC4a increases LeEIX2 presence on endosomes under basal conditions (Supplementary Fig. 5a), and the cells respond further to EIX with an increase in LeEIX2 endosomal presence. Interestingly, overexpression of the 67 amino acid peptide increases LeEIX2 presence on endosomes similar to the full-length peptide under basal conditions, however, in accordance with the MAMP-independent responses we observed in the *slnrc4a* mutant, the cellular response to EIX is

diminished. Protein expression of the 67 amino acid peptide under transient conditions is similar to the Free-mCherry control, while the full length SlNRC4a protein is expressed at lower levels (Supplementary Fig. 5b). Interestingly, treatment with the MAMP EIX increased the level of the Full SlNRC4a protein but not of the 67 amino acid peptide (Supplementary Fig. 5b), as we previously demonstrated[19]. This result corresponds with those obtained under basal conditions in the *slnrc4a* mutant, suggesting that the 67 amino acid peptide does not require MAMP/ligand stimulation to mediate some, though not all, of its effects.

## Discussion

Plants defend themselves against potential pathogens via an innate immune system. Here, we have found that upregulating the innate immune system by genetically enhancing the function of *SlNRC4a* can increase both MAMP-mediated and MAMP independent defense responses, resulting in a stronger immune system output and disease resistance. Systemic plant immune responses can be sensitized by priming, where plants potentiate defensive capacity after sensing diverse stimuli[58]. Priming reportedly results in faster, stronger and/or more sustainable defense responses[59], making it an agricultural strategy that can be used by farmers to improve crop resistance. Our work suggest that genetically enhancing innate immunity may be agriculturally superior to priming strategies, with an upregulation in defense responses free of the requirement for priming agents, leading to better reproducibility. *slnrc4a* gain of function mutant lines display higher levels of ethylene, callose deposition, pathogen responsive gene expression, PRR gene expression, glandular trichome head size, and secondary metabolite emission in steady state conditions (Figs. 2 and 3). Many of the reported mechanisms underlying the defensive functions of trichomes in the Solanaceae are Jasmonic acid (JA) dependent. Upregulation or treatment with JA can increase the abundance of trichomes, both glandular and non-glandular, as well as activating plant defensive pathways, leading to increased pest and pathogen resistance[60–64]. In some cases, activation of JA dependent pathways does not lead

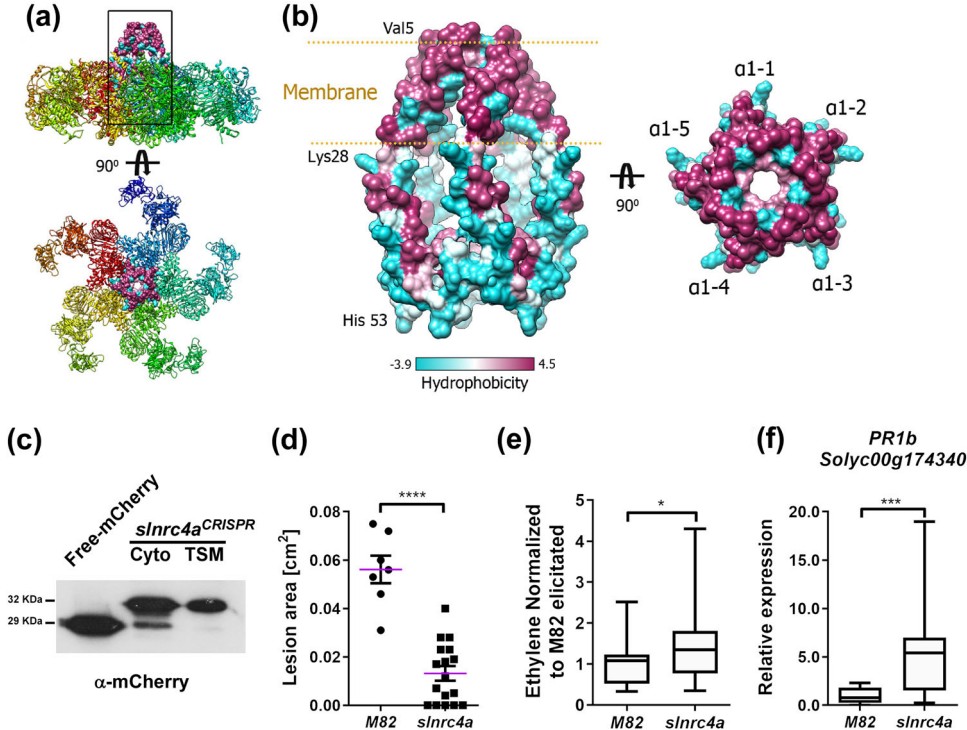

**Fig. 7 slnrc4a can reside in the membrane and does not require PRR activation. a** The complex structure of ZAR1 (PDB:6J5T) represented as a ribbon in which each distinct color represents a different protein chain. The square box marks the position of the derived peptide in the surface representation. **b** Left, the modeled 3D assembly of SINRC4a$^{1-67}$ peptide based on the ZAR1 structure, in surface representation colored by hydrophobicity. The predicted membrane insertion site is marked by the two gold lines. Right, a top view of the SINRC4a$^{1-67}$ peptide structure with the marked five α1 helices (α1,1-5). **c** Biochemical protein fractionation of SINRC4a$^{1-67}$-mCherry transiently expressed in *N. benthamiana*. Cytosolic (Cyto) or Triton X-100 soluble membrane (TSM) protein fractions were subjected to SDS-PAGE followed by α-mCherry immunoblot. Cytosolic fraction of free-mCherry transiently expressed in *N. benthamiana* is shown. **d** M82 and *slnrc4a* tomato lines were grown in-vitro for 3-weeks, leaves were inoculated with *B. cinerea* (10X$^6$ spores/mL) and lesion area was measured after 3 days. Individual values are all presented line indicates mean. **e** Ethylene induction after EIX elicitation in M82 and *slnrc4a* samples was measured using gas chromatography. M82 average ethylene production after elicitation is defined as 1, average ± SEM is presented (N$_{M82}$ = 17 and N$_{slnrc4a}$ = 29). Asterisks represent statistical significance in *t*-test with welch correction (**\*\*p*-value < 0.01). **f** M82 and *slnrc4a* tomato lines were grown in-vitro for 3-weeks and PR1b expression was measured by RT-qPCR in steady state conditions. Relative expression normalized to M82 (mock). For **d**, **f** 4–8 individuals were analyzed, average ± SEM is shown. Statistical significance in *t*-test with welch correction (**\*\*\*p*-value < 0.001). **e**, **f** Boxplots are shown with inter-quartile-ranges (box), medians (black line in box) and outer quartile whiskers, minimum to maximum values.

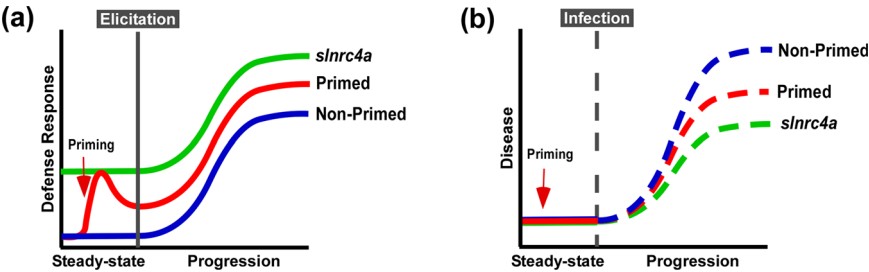

**Fig. 8 Representative model of MAMP-elicitation and disease response. a** Plants deploy defense responses after MAMP-elicitation (blue), which can be enhanced by priming, triggering a stronger and/or faster response (red). We propose that in *slnrc4*, plant defense responses are pre-activated, and this leads to stronger responses to MAMP-elicitation (green). **b** Plant disease progression in natural conditions (blue), can be retarded when plants are pre-treated with priming agents (red). We propose that in *slnrc4*, disease progression is lower due its pre-activated basal condition, in a manner which may be superior to primed plants (green).

to an increase in trichome density, but causes upregulation of metabolites[65] or an increase in plant defense responses[66,67] which is sufficient to increase resistance. However, though several reports indicate that the amount of defensive metabolites produced is a function of the trichome head size[68–70], as we report here, few works have related directly to the connection between trichome head size and pathogen resistance or plant defense pathways. Although most reports concerning the connection

between trichomes and defense point to JA dependent mechanisms, JA and SA activated pathways are not necessarily discrete in tomato[46,47,66].

Interestingly, in *slnrc4a* lines, the upregulation of basal immunity parameters is constitutive, and they do not return to basal levels in the absence of pathogen attack, as occurs in classical priming in WT plants (see also Fig. 8). *slnrc4a* exhibited higher MAMP-mediated defense responses and enhanced biotic

resistance (Fig. 1, Supplementary Fig. 2), suggesting that *slnrc4a* mutated lines are genetically immuno-activated.

An increase in defense parameters under basal conditions has been described for plants overexpressing NPR1 and AP2/ERF as well as for *cpr1-1*, a MAPKKK mutant[71]. The risk of constitutively activated immunity lies in compromising plant fitness and/or yield, due to allocation of resources and/or accumulation of toxic defense compounds[56,71]. The *slnrc4a* gain of function mutation appears to have no cost on plant fitness in greenhouse and lab-in-field conditions, presenting an important agricultural advantage when compared to plants with constitutively active defense. We have not examined the performance of *slnrc4a* plants under variable stress conditions. This will be investigated agriculturally in future research. Figure 8 depicts a model representing elicitation (a) and disease (b) states in the *slnrc4* gain of function mutants in comparison with primed and non-primed WT plants.

Here we report that two nearly identical truncated peptides, derived from two *SlNRC4a* Cas9 generated gain of function mutants, both induce immunity to greater levels than the WT protein. In several cases, truncated versions of NLR proteins, consisting of the CC domain (CCd) or even only the first 29 aa that form an N-terminal a-helix, were able to induce spontaneous ligand-independent HR, although longer truncated versions that include the 29 aa did not elicit ligand-independent HR[22], possibly due to inhibitory modules within the larger NLR negatively regulating spontaneous HR. The *slnrc4a* truncated gain of function mutant in this work, which does not induce ligand independent HR, consists of 67 N-terminal amino acids, which encompass the region of 29 aa in the α-helix reported to induce autonomous HR[22]. *slnrc4a* possesses upregulated ethylene, callose deposits, secondary metabolites and defense gene expression under steady state conditions (Fig. 2), with no changes in ROS, electrolyte leakage and HR parameters in steady state. Oxidative burst and electrolyte leakage are potent responses, resulting in PCD/HR cell death[31]. This demonstrates the tight control of plant defense, affording enhanced immunity through upregulation of steady state non-lethal defense responses, while avoiding unnecessary upregulation of lethal ones. This delicate balance of ligand-dependent regulated processes and constitutively active basal immunity enables the elevated tolerance to biotic stress without noticeable cost on fitness and yield.

Recently, the structure of ZAR1, a CNL-type NLR, has been elucidated, demonstrating the formation of a pentameric α-helical barrel structure that interacts with the pseudo-kinase RKS2 to form the sensor complex, termed the resistosome[13,14]. The complex structure, which likely resides in an inactive state, is re-shaped after ligand activation, exposing the N-terminal α-helix proposed to form a plasma membrane pore that triggers PCD[13,14]. Negative regulation of immune signaling has been reported in many cases[16,72,73]. SlNRC4 possesses sufficient homology to ZAR1 to possibly participate in a resistosome structure, as our predicted structure suggests (Fig. 7). Our data indicate that the truncated N-terminal 67 amino acid SlNRC4 protein is able to associate with the membrane similar to the full length protein (Fig. 7). Due to its size, it is likely unable to form most protein–protein interactions in which the full length protein is involved, suggesting that these interactions normally block defense responses. Additionally, inhibitory activity of the full length NRC4a on itself could prevent activation of defense mediated by the short NRC4 peptides, both the 29 aa peptide[22] and the 67 aa peptide investigated here. Since the inhibitory interactions formed by the full length SlNRC4 are mostly absent in our *slnrc4a* mutant lines, the partial inhibition achieved leads to a steady state enhancement of defense. A shorter NRC4 N-terminal peptide comprised of 29 amino acids has been reported

to cause constitutive HR[22], likely due to even stronger de-regulation of inhibitory interactions within NRC4a, or with additional proteins.

Our results suggest that *slnrc4a* broad-spectrum biotic resistance is likely promoted by increased basal immunity. This could suggest that SlNRC4a has a role in immune responses prior to or independent of resistosome activation. NRC4 has been shown to facilitate both MAMPs and/or effector signal transduction[12,19,20]. The 67 aa truncated peptide in the *slnrc4* gain of function mutant likely does not require MAMP/effector signaling to mediate some of its effects. This is in line with the activated basal immunity parameters in the *slnrc4* mutant, which were retained under sterile conditions, and the inability of EIX to affect trafficking of LeEIX2 when the 67 aa peptide is expressed, strongly indicating that some aspects of the *slnrc4a* mutant phenotype are MAMP independent. It was recently reported that cell damage results in upregulation of PRRs in neighboring cells, suggesting that localized immune responses rely primarily on PRR presence[74]. Therefore, the increased basal expression of PRRs in slnrc4a could explain the upregulation of MAMP-independent responses. MAMP-independent increased basal immunity in *slnrc4a* increases defense responses and diseases resistance.

We previously reported that silencing members of the NRC4 clade caused a significant decrease in EIX mediated defense responses[19]. A recent report that deletion of the SlNRC4 gene cluster compromises Rpi-blb2 but not flg22-induced responses (Wu et al.,[20]), prompted us to examine this clade-deletion mutant and compare it with our gain of function mutant. The results (Fig. 6) confirm that, with respect to the fungal MAMP EIX, the NRC4 clade deletion has reduced defense responses, similarly to the silenced lines we previously reported[19], with differences in amplitude of defense responses likely stemming from the differences in background cultivar and silencing methodology. On the other hand, the mutation retaining the 67 amino acid peptide of NRC4a we characterize here, behaves as a gain of function, increasing disease resistance. These results indicate that SlNRC4a is differentially involved in MAMP mediated responses, and together with the wide array of NRC4 mutants displaying different defense phenotypes, provide interesting prospects for future research.

## Methods

**Plant materials and growth conditions**. *Solanum lycopersicum* cv M82 and homozygous T3 and T4 *slnrc4a* independent CRISPR lines *slnrc4a-2* and *slnrc4a-5* (previously described in ref.[19]) were grown from seeds in soil (Green Mix; Even-Ari, Ashdod, Israel) in a growth chamber, under long-day conditions (16 h:8 h, light:dark) at 24 °C. Plants from both independent CRISPR lines were used in all assays.

For experiments in sterile conditions, tomato plants were grown in 0.5X MS media (without sucrose supplementation) in magenta boxes or in sterile glass test tubes with valve lids for ethylene assays.

**Bacterial infection**. *Pseudomonas syringae* pv tomato (strain DC3000) and *Xanthomomas campestris* pv Vesicatoria (strain 85–10) were used for bacterial infection analysis. Bacterial cultures were grown in LB medium containing 100 mg L$^{-1}$ of rifampicin (for both) and 300 mg L$^{-1}$ of streptomycin (for *X. campestris*) overnight at 28 °C. Bacterial cultures were centrifuged and resuspended in 10 mM MgCl$_2$ for a final concentration of 10X$^5$ CFU/mL (OD$_{600}$ = 0.0002). Six-week-old tomato plants were vacuum immersed with the bacterial suspensions. Three days after infiltration, three-leaf discs of 0.9 cm diameter were sampled from leaves 4–6 of at least four plants from each genotype and ground in 1 mL of 10 mM MgCl$_2$. Bacterial CFU were determined by plating 10 μL from 10-fold serial dilutions and counting the resulting colonies. Negative controls consisted of 10 mM MgCl$_2$ without pathogen inoculation. For equal bacterial loading verification control, leaf discs harvested 4 h after infiltration were used. The experiments were replicated three independent times using three individuals for each genotype and four technical replicates ($N_{total}$ = 24).

**Fungal infection**. Pure *B. cinerea* (Bc16) and *S. sclerotiorum* (Scl5) cultures were grown on potato dextrose agar (PDA) (Difco Lab) plates and incubated at 22 °C for

5–7 days. *B. cinerea* spores were harvested in 1 mg ml$^{-1}$ glucose and 1 mg ml$^{-1}$ $K_2HPO_4$ and filtered through cheesecloth. Spore concentration was adjusted to $10^6$ spores ml$^{-1}$ using a haemocytometer. Leaves 4–6 from 5 to 6-week old tomato plants, or leaves from in-vitro grown 3-week old plants for sterile experiments (Fig. 7), were excised and immediately placed in humid chambers. Each tomato leaflet was inoculated with two droplets of 10 µL spore suspension. For *S. sclerotiorum*, uniform mycelial plugs (5 mm diameter) were taken using a cork-borer from colony margins and placed mycelial side down on the adaxial surface of each leaf. Inoculated leaves were kept in a humid growth chamber at 21 °C. *B. cinerea* experiment was replicated four independent times, using leaves from five individuals for each genotype and 3–4 technical replicates ($N_{total} = 70$). *S. sclerotium* experiment was replicated three independent times using leaves from three individuals for each genotype and two technical replicates ($N_{total} = 18$).

*O. neolycopersici* was isolated from young leaves of 4–6 week old tomato plants grown in a commercial greenhouse in the winter of 2019. Conidia of the pathogen were collected by rinsing infected leaves with sterile water. For the artificial infection of tomato leaves, the concentrations of these conidial suspensions were determined under a light microscope using a hemacytometer. All suspensions were adjusted to $10^4$ ml$^{-1}$ and then sprayed onto 5–6 week old tomato plants at a rate of 5 ml per plant. All suspensions were sprayed within 10–15 min of the initial conidia collection. Suspensions were applied with a hand-held spray bottle and plants were left to dry in an open greenhouse for up to 30 min. Inoculated leaves were kept in a humid growth chamber at 21 °C. The experiment was replicated three independent times using three individuals for each genotype and 2–3 technical replicates ($N_{total} = 21$).

Controls consisted of leaves treated with water/buffer without the inoculation of pathogen. The diameter of the necrotic lesions or % of infected leaf tissue was measured three to four days post inoculation, as indicated and calculated using ImageJ.

**Pest quantification**. *Tuta absoluta* and *Bemicia tabaci* assays were performed in a thoroughly infested chamber by removing pest maintaining plants and randomly placing 4–5 week old assayed plants. After 2 weeks, plants were assessed for disease levels. For *T. absoluta*, all leaf tissue was removed and % of infected leaf tissue was calculated. For *B.tabaci*, leaflets from leaf number 4–6 were gently removed and placed in poly bags. Bags were placed in the freezer. *B. tabaci* numbers were subsequently counted under a stereo microscope. The experiments were replicated three independent times using three individuals for each genotype and two technical replicates ($N_{total} = 18$).

**Oxidative burst measurement**. ROS measurement as previously described[75]. Leaf disks 0.5 cm in diameter were taken from leaves 4 to 6 of 5–6 week old *M82 and slnrc4a* tomato leaves. Disks were floated in a white 96-well plate (SPL Life Sciences, Korea) containing 250 µl distilled water for 4–6 h at room temperature. After incubation, water was removed and ROS measurement reaction containing either 1 µg/mL EIX or water (mock) was added. Light emission was immediately measured using a luminometer (Turner BioSystems Veritas, California, USA). Each experiment was repeated four times with 12 technical replicates ($N_{total} = 48$).

**Ethylene measurement**. Ethylene production was measured as previously described[75]. Leaf disks 0.9 cm in diameter were taken from leaves 4 to 6 of 5–6 week old *M82 and slnrc4a* tomato leaves. Disks were washed in water for 1–2 h for EIX and steady state assays or incubated for 3–4 h in 1 mg ml$^{-1}$ glucose and 1 mg ml$^{-1}$ $K_2HPO_4$ (with or without *B. cinerea*) for *B. cinerea* assays. Every six disks were sealed in a 10 mL flask containing 1 ml assay medium (with or without 1 µg/mL EIX or with or without *B. cinerea*) for 4 h (for EIX) or overnight (for *B. cinerea*) at room temperature. Ethylene production was measured by gas chromatography (Varian 3350, Varian, California, USA). The experiment was replicated five independent times using a pool of leaves from 4 to 6 individuals and 10 technical replicates ($N_{total} = 50$). In the case of sterile experiments, ethylene was measured 24 h after sealing of individually grown 3-week old plants in glass test tubes ($N_{M82} = 17$, $N_{slnrc4a-2} = 16$ and $N_{slnrc4a-5} = 13$).

**Electrolyte leakage measurement**. Leaf disks 0.9 cm in diameter were taken from leaves 4 to 6 of 5–6 week old M82 and *slnrc4a* tomato leaves. Disks were washed in a 50 mL water tube for 3 h. Every five disks were floated in a 12-well plate containing 1 mL of water with or without 1 µg/mL EIX (adaxial surface down) at room temperature with agitation. Conductivity was measured in the water solution after 40 h incubation using an electrolytic conductivity meter (EUTECH instrument con510). The experiment was replicated four independent times using a pool of leaves from 4 to 6 individuals and 8 technical replicates ($N_{total} = 32$).

**Callose deposition**. Leaves 4–6 of 5–6 week old M82 and *slnrc4a* plants were pre-treated with 1 µg/mL EIX or water (mock) for 24 h through petiole before harvesting leaf disks 0.9 cm in diameter. Samples were cleared by rinsing in ethanol 50% and then stained with aniline blue 0.01% (150 mM $K_2HPO_4$ pH 9.6) for 30 min in vacuum. Samples were mounted on glycerol 50% and images were taken using a Zeiss LSM780 confocal microscope system with EC Plan-Neofluar 10X/0.37 M27 objective and tiling function activated in order to register the whole leaf disc.

The excitation wavelength used was 405 nm (1% power) and the emission was collected in the range of 473–532 nm. Images of 8 bits and $1024 \times 1024$ were acquired using a pixel dwell time of 1.58, pixel averaging of 4 and pinhole of 2 airy units. Image analysis was performed using Fiji-ImageJ using the 3D Object counter tool[76]. Three independent replicates of 4 leaf discs ($N = 12$) were analyzed. The experiment was replicated three independent times using leaves from three individuals for each genotype and two experimental replicates ($N_{total} = 18$).

**RNA extraction and qRT-PCR analysis**. Plant total RNA was extracted, from leaves of 4–5-week old plants or from leaves of 3-week old plants grown in-vitro, using SV Total RNA Isolation System (Promega, Madison, WI, United States). Four µg RNA samples were subjected to first strand cDNA synthesis using M-MLV reverse transcriptase (Promega, Madison, WI, United States) and oligodT$_{15}$. RT-qPCR was performed according to the Fast SYBR Green Master Mix protocol (Life Technologies, Thermo Fisher, Waltham, MA, United States), using a StepOnePlus machine (Thermo Fisher, Waltham, MA, United States). Specific primers used in this work are detailed in Supplementary Table 2. Relative expression quantification was calculated using the Pffafl method[77] for EIX experiments and copy number method for *B. cinerea* experiments[78]. Sl-cyclophilin (Solyc01g111170) was used as reference gene in all analyses[19]. The experiment was replicated 3 independent times, using 4 individuals for each genotype and performing technical triplicates ($N_{total} = 24$).

**Metabolomic analyses**. Frozen powder (1g) from leaves 4–6 of 5–6 week old tomato plants was homogenized in saturated NaCl solution (3 mL), and 0.3 µg of internal standard (2-heptanone) and NaCl powder (1g- to inhibit enzymatic reactions) were added. Samples were sealed in 20 ml vials and kept at 4 °C until analysis. Vials containing 20% NaCl were used as control. Samples were preheated to 45 °C for 10 min, then the fiber was inserted into the vial's headspace. After 25 min, SPME syringe was introduced by automatic HS-SPME MPS2 (Gerstel, Mülheim, Germany), at 50 °C by 65 µm into the injector port of the GC-MS apparatus for 5 min in a splitless mode.

SPME fiber was injected into an Agilent GC 6890 system, coupled to quadrupole mass spectrometer detector 5973N (CA, USA). The instrument was equipped with Rxi-5sil MS column (30 m length × 0.25 mm i.d., 0.25 µm film thickness, stationary phase 95% dimethyl- 5% diphenyl polysiloxane). Helium (13.8 psi) was used as a carrier gas with splitless injection. The identification of the volatiles was assigned by comparison of their retention indexes with those of literature and by comparison of spectral data with standard or with the Nist 98 and QuadLib 2205 GC-MS libraries. Component amount in each sample was calculated as (peak area x internal standard response factor) divided by (response factor x internal standard peak area). The results are an average of three biological replicates and presented as ng compound/gr tissue/vial. The experiment was replicated 3 independent times using a pool of 4 individuals for each genotype and two technical replicates ($N_{total} = 6$).

**Trichome morphology**. For SEM analysis of leaf trichomes, samples from 8 different 5–6-week old plants (the right-hand middle leaflet of leaf 5 or 6) were excised and fixed using the methanol method[79]. The samples were then critical-point-dried in a Critical Point Dryer (CPD-030, Bal-Tec/Leica) and glued on stubs. The samples were gold coated in a Gold Sputter Coating Unit (Quorum Technologies/Polaron, UK) and observed by Low Vacuum SEM (JSM 5410 LV, Jeol Ltd., Japan). The experiment was replicated three independent times, using leaves from three individuals for each genotype ($N_{total} = 18$).

**Growth measurement of tomato**. Growth measurement were performed as previously described by Gur et al.[80]. Red and green fruits were counted and weighed separately. Plant vegetative weight was determined by weighing only the vegetative tissue (after harvesting the fruits) without the roots. Total fruit yield per plant included both the red and the green fruits. Concentrations of total soluble sugars were measured as BRIX percentage on a digital refractometer with a range of BRIX 0–85% ± 0.2%, from a random sample of 5 red fruits per plant. Harvest index (HI) was calculated as the ratio between the total yield and total biomass. The experiment was replicated three independent times, using six individuals for each genotype ($N_{total} = 18$ per genotype).

**Structural analysis of NRC4a derived peptide**. The 67 aa peptide from NRC4a tertiary structure was modeled on Arabidopsis thaliana ZAR1 (PDB: 6J5T.A)[13] with SWISS-MODEL—https://swissmodel.expasy.org[81]. Modeled tertiary structure of NRC4a were superimposed on the five monomers of ZAR1 complex to create the quaternary motamer using Coot[82]. To ensure good complex, wrong rotamer in Phe8 that led to side chain collision was replaced with other rotamer. The formed structure was energetically minimized using swiss PDB viewer (http://www.expasy.org/spdbv/)[83]. Figures were made using UCSF Chimera[84].

**Cellular fractionation and western blotting**. Western blot was performed on *N. benthamiana* leaves transiently expressing slnrc4a$^{CRISPR}$ constructs[19]. One hundred milligrams of plant tissue were ground with liquid nitrogen, and three

volumes of extraction buffer (50 mM of Tris-HCl, pH 7.5, 2 mM of MgCl2, 150 mM of NaCl, 140 mM of β-mercaptoethanol, 2 mM of phenylmethylsulfonyl fluoride (PMSF), and one complete protease inhibitor tablet, without EDTA [Roche, Germany] per 50 ml) were added. Samples were centrifuged, and supernatant cytosolic fraction was discarded (or collected if required). Pellets were ground using two volumes of EB with 1% Triton X-100 and incubated in a rotating wheel at 4 °C for 20 min before centrifugation. Supernatant samples (TSM) were collected and boiled after adding sample buffer (8% sodium dodecyl sulfate, 40% glycerol, 200 mM of Tris-Cl, pH 6.8, 388 mM of dithiothreitol, and 0.1 mg ml$^{-1}$ of bromophenol blue dye). Samples were run in sodium dodecyl sulfate-polyacrylamide gel electrophoresis, blotted onto nitrocellulose membranes, and incubated rat anti-mCherry (Chromotek).

**Statistics and reproducibility**. All data are presented as average ± SEM. Differences between two groups were analyzed for statistical significance using a two-tailed $t$-test. Differences among three groups or more were analyzed for statistical significance with a one-way ANOVA. Regular ANOVA was used for groups with equal variances, and Welch's ANOVA for groups with unequal variances. When a significant result for a group in an ANOVA was returned, significance in differences between the means of different samples in the group were assessed using a post-hoc test. Tukey was employed for samples with equal variances when the mean of each sample was compared to the mean of every other sample. Bonferroni was employed for samples with equal variances when the mean of each sample was compared to the mean of a control sample. Dunnett was employed for samples with unequal variances. All statistical analyses were conducted using Prism8$^{TM}$. All experiments were conducted in at least three biologically independent repeats. The number of replicates is indicated for each experiment in each figure legend and corresponding method section.

**Reporting summary**. Further information on research design is available in the Nature Research Reporting Summary linked to this article.

## Data availability
The authors declare that the data supporting the findings of this study are available within the paper and its Supplementary information files. Raw data is available from the corresponding author upon reasonable request.

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

## Acknowledgements

The authors thank Yigal Elad for providing *B.cinerea* and *S. sclerotiorum* strains, Guido Sessa for providing the *Xcv* strain, and Sophien Kamoun for providing seeds of the tomato NRC4 near-full clade deletion mutant. This work was partly supported by the Israel Science Foundation administered by the Israel Academy of Science and Humanities, Grant No. 550/18. RG is supported by the Indo-China ARO Postdoctoral Fellowship Program. MB thanks members of the Bar group for continuous discussion and support. Publication No. 600/19 of the ARO.

## Author contributions

M.B., A.A., M.L.-M., and L.P. conceived and designed the study. L.P., M.L.-M., R.G., I.S., N.K., R.Z., and R.D.-R. formulated the methodology and carried out the experiments. M.L.-M., L.P., E.L., R.D.-R., E.B., R.Z., A.A., and M.B. analyzed the data. All authors contributed to the writing of the paper.

## Competing interests

The authors declare no competing interests.
