## [Peer Review File · Communications Biology]

Reviewers' comments:

Reviewer #1 (Remarks to the Author):

The manuscript by Pizarro et al. reported a specific gain of function mutation in a helper NLR protein, SINRC4a, in tomato resulted broad spectrum resistance to diverse pathogens. The developed genome edited slnrc4 mutants encoding a 67 amino acid truncated protein displayed strong response to the elicitors of flg22 and EIX and showed defense resistance to bacteria, fungi, and insects transmitted viral diseases (pests). Constitutive priming of slnrc4 was observed on ethylene production, callose deposition, and defense related marker genes expression, elevated defense metabolites and increased trichome head size. Compared with the control, the mutants were found to have enhanced defense response upon elicitation with ROS burst, dramatically increasing of ethylene production and callose deposition. The increased disease resistance was evidenced by the up-regulation of pathogen response gene expressions. Interestingly, no significant yield tradeoff with the enhanced resistance was found for the slnrc4 mutants which is most valuable for tomato breeding.

I have only two minor concerns regarding to the current manuscript:

1. The deletion of NRC4 gene cluster has significant reduction in immunity revealed the loss of function of the helper NLR. SINRC4a functions as interacting partner for PRRs in the membrane that is mostly involved in ETI. The truncated 67 aa protein of SINRC4a still worked in the membrane and involved in the NRC4 conformation. However, broad-spectrum resistance to an array of pathogens is independent of the eliciting of pathogens that involved in PTI. The transition the slnrc4a function indicated a fine-tuning of the helper protein in plant defense response. Further discussion may be addressed to compare the differential response of the 29 aa, 67 aa truncated protein and the entire deletion.
2. It is difficult to understand the constitutive priming of defense response with elevated ethylene production, callose deposition, PR genes expression and defense metabolites with significant yield cost. The mutants must have a strong metabolic ability to maintain high levels of gene expression in these pathways. Four traits were evaluated for the mutants in the green house. Is there any other agronomic traits influenced?

Reviewer #2 (Remarks to the Author):

This manuscript entitled 'A gain of function mutation in SINRC4a primes broad spectrum disease resistance' by Pizarro et al shows that a mutation in tomato helper NLR, NRC4a, resulted in conferring disease resistance to several types of pathogens and pests. The mutation in NRC4a was previously reported. Importantly, slnrc4a mutant described in this manuscript shows enhanced defense responses such as transcriptional and metabolic changes.

The enhanced disease resistance of slnrc4a mutant to multiple pathogens is interesting observation. However, this manuscript lacks mechanistic investigation. In particular, there is no clear demonstration on why the short variant of NRC4a confers enhanced defense responses and pathogen resistance. In slnrc4a mutant, what is the mRNA and protein levels of NRC4a mutant variant? Has the mutation caused an increased mRNA or protein stability? Does the short variant interferes or activates with other NRC or NLR variant function? It was shown by the same group that SINRC4, in particular CC domain, physically associates with PRR. Does the short form (a part of CC domain) of SINRC4 associate with LeEIX2 or FLS2? Unfortunately, without designing of experiments to elucidate the mechanistic basis of enhanced resistance conferred by SINRC4a truncated variant, the current version

of manuscript seems to be descriptive and speculative.

Reviewer #3 (Remarks to the Author):

A gain of function mutation in SINRC4a primes broad spectrum disease resistance.

Pizarro et al

Nat comm

In this interesting study the authors further their investigation of a truncated NRC4a allele. In a previous study (PCE 2018) the authors generated Crispr/cas9 truncation mutants of SINRC4a and found that these plants showed an enhanced defense response to the EIX elicitor and increased resistance to infection of *Botrytis cinerea*. Here, they show that these plants exert heightened resistance to a range of pathogens including bacteria (*Pst* and *Xcv*), bio- and necrotrophic fungi (*Bc* and *Sc*) and two pests (whitefly and tomato leaf miner). Increased resistance correlates with induction of defense markers such as ethylene, defense gene (PR genes and NRCs) expression, terpenoid production and callose depositions, but not ROS production or ion leakage. Upon MAMP treatment or *B. cinerea* infection the *slnrc4* mutants show a potentiated defense response including heightened ROS burst, ethylene production and callose deposition. Surprisingly, the *slnrc4* mutants do not seem to exert pleiotropic phenotypes as their yield in optimal lab/greenhouse conditions are not different than those of wild type plants. Biochemical fractionation of the putative NRC4 fragment produced in the *slnrc4* mutant indicates that it is membrane localized.

Based on the presented data the authors propose that NRC4 mutants might be applicable in agricultural setting to confer broad spectrum resistance to plant pathogens without negatively affecting yield.

The paper is well written and the conclusions are based on a large set of high-quality experiments. Nevertheless, I have some comments for consideration.

Major comments:

- The authors claim that the *slnrc4* mutant is a gain-of-function mutation. The proposedly new function that the authors assign to it is that it causes a constitutive priming phenotype explaining the increased resistance to various pathogens and pests. I do not agree with this concept. The formal definition used in the community for priming is that plants are capable to enhance their basal defense strategies against harmful organisms upon the perception of certain stimuli. This enhanced resistance is not necessarily accompanied by direct activation of defenses, but often depends on a sensitization of the plant tissue to express defenses faster and/or stronger when the plant is under attack. As the *slnrc4* mutants clearly show enhanced basal levels of defence activation in the absence of any pathogen or biotic stress actor this finding disqualifies the phenotype as being primed. Indeed, not the full spectrum of defence outputs is activated (e.g. no ROS, not ion leakage, no cell) resulting in the absence of any morphological phenotypes when grown under optimal condition. As the plants are sensitized and do respond faster upon pathogen/MAMP perception some sort of priming might also be induced in these plants, but in addition they exert an autoimmune phenotype. I would suggest the authors to adjust their claims in abstract, into an discussion that states that the mutant induces a priming phenotype.

- Two *slnrc4* mutants have been generated that differ in a SNP in the coding sequence resulting in the induction of a stop codon. In the manuscript it is not clear whether the mutant shows the exact same phenotypes and which mutant is used in which assay. It would strengthen the study if the authors can show both mutants in all panels as if they give the same output in all cases the phenotype is likely to

be caused by the mutation in *nlrc4* and not by another linked mutation in the genome. Are both mutants derived from an independent transformation, or do they originate from the same callus/line and hence are not truly independent.

- L133-149 The authors monitor expression of a number of defence marker genes such as PR proteins that are proxies for JA, SA and or ISR/SAR. It would strengthen the study if also expression of the PRR FLS2 would be monitored as defence induction typically also correlates with induced accumulation and concomitant expression of PRRs. Information about PRR FLS2 expression (together with that of EKS) would provide insight whether the observed resistance is due to a mild autoimmune phenotype and heightened PRR expression or try priming, in which these receptors are not induced.

- L166-170, an increased size of glandular trichomes is observed. This aspect is not discussed further in the manuscript. Is this a phenotype more often seen in autoimmune phenotypes and or during priming. Please elaborate on this unexpected finding in the discussion of the manuscript.

- L203, please show the data to support the claim that plants showing more severe disease symptoms show higher expression of the defence marker genes.

- L215-225, the claim that the plants are gmo free does not correspond with current legislation in e.g. Europe as the mutants have been generated using a technology that uses genetic modification. Please correct this paragraph and related section into that *slnrc4* plants that do not contain transgenes retain the phenotype. Consider to mention that these results confirm that the observed phenotype are not due to a mutation due to insertion of the T-DNA. As it is unclear how the mutants have been made this could be an alternative explanation for the observed findings.

- L247 states: "As we demonstrated here that the 67 amino acid CRISPR generated peptide induces a constitutively primed state". Besides that priming is not shown it is also not shown that the 67 aa protein is stably produced and accumulates in the mutants. Please tune down this claim, or show accumulation of the peptide (or at least expression of the mRNA) in the mutant lines.

- L249, "we proceeded to examine whether this peptide retains membranal localization". To me it is unclear where "retains" refers to. What are the assumptions that a membrane localization is foreseen? Please provide reference and explain, or rephrase.

- and L251, "showing that it retain the ability to..." is overinterpretation of the data. That the protein can be modeled on the ZAR1 structure does not mean that it has the properties and has the same confirmation. Please rephrase/remove.

- L254, also in planta grown plants can experience stress and perceive environmental stimuli. Consider rephrasing this part, in e.g. are not exposed to biotic stresses or to MAMPs.

- L303-311, an equal likely explanation for the observed defense in the ability to of the longer and shorter CC fragment to trigger immune activation is that the C-terminal part of the CC fragment has an autoinhibitory activity. This would not require any additional protein-protein interactions and its removal would potentiate the immune response resulting in pcd.

Minor:

L 63, citation Ron and Avni incomplete

L124 remove "."

L184-186, Overinterpretation of the data. A correlation between enhanced defense responses and increased resistance is not the same as a causal relation. See also above my remark about priming.

L188, "priming can result in improved disease outcomes". Please adjust, in e.g. in priming can result in improved resistance, or priming can result in decreased disease outcomes.

L246, remove d in and.

Fig S1, include in these graphs also the non EIX treated control for comparison.

L606, include journal.

Many thanks to the reviewers for their time. We believe the comments have greatly improved the manuscript. Follows a "point by point" response; all line numbers referenced relate to the marked up version of the manuscript.

Reviewer #1:

"The manuscript by Pizarro et al. reported a specific gain of function mutation in a helper NLR protein, SINRC4a, in tomato resulted broad spectrum resistance to diverse pathogens. The developed genome edited slnrc4 mutants encoding a 67 amino acid truncated protein displayed strong response to the elicitors of flg22 and EIX and showed defense resistance to bacteria, fungi, and insects transmitted viral diseases (pests). Constitutive priming of slnrc4 was observed on ethylene production, callose deposition, and defense related marker genes expression, elevated defense metabolites and increased trichome head size. Compared with the control, the mutants were found to have enhanced defense response upon elicitation with ROS burst, dramatically increasing of ethylene production and callose deposition. The increased disease resistance was evidenced by the up-regulation of pathogen response gene expressions. Interestingly, no significant yield tradeoff with the enhanced resistance was found for the slnrc4 mutants which is most valuable for tomato breeding.

I have only two minor concerns regarding to the current manuscript:

1. The deletion of NRC4 gene cluster has significant reduction in immunity revealed the loss of function of the helper NLR. SINRC4a functions as interacting partner for PRRs in the membrane that is mostly involved in ETI. The truncated 67 aa protein of SINRC4a still worked in the membrane and involved in the NRC4 conformation. However, broad-spectrum resistance to an array of pathogens is independent of the eliciting of pathogens that involved in PTI. The transition the sinrc4a function indicated a fine-tuning of the helper protein in plant defense response. Further discussion may be addressed to compare the differential response of the 29 aa, 67 aa truncated protein and the entire deletion."

Many thanks for this comment. We added more discussion of this issue to the manuscript (e.g., lines 347-354).

"2. It is difficult to understand the constitutive priming of defense response with elevated ethylene production, callose deposition, PR genes expression and defense metabolites with significant yield cost. The mutants must have a strong metabolic ability to maintain high levels of gene expression in these pathways. Four traits were evaluated for the mutants in the green house. Is there any other agronomic traits influenced?"

To address this, we amended Figure 5 to include additional traits that were examined. Additionally, we have added some discussion in the text of this- as while the behavior of these mutant lines is the same under optimal conditions, we have not examined them under various stress conditions, which is the subject of future work (lines 319-321). New figure 5 is attached here for convenience.

Figure 5. *slnrc4a* gain of function mutant line has similar yield production to WT. Phenotypic parameters of yield were measured in M82 and *slnrc4a* plants. **(a)** Total fruit number is presented as number of tomato fruits produced by plant. **(b)** Total fruit weight per plant was measured. **(c-d)** same as a-b, but ripe fruit only. **(e)** Total plant fresh weight (aerial tissues only). **(f)** Harvest Index (HI) of plants was calculated as the ratio between the total fruit yield mass and total biomass. **(g)** Total soluble sugars were measured by refractometry expressed as °Brix. Average \pm SEM of 3 independent replicates is shown. No statistically significant differences were observed among WT and *slnrc4a* (t-test, Welch's correction).

Reviewer #2:

"This manuscript entitled 'A gain of function mutation in SINRC4a primes broad spectrum disease resistance' by Pizarro et al shows that a mutation in tomato helper NLR, NRC4a, resulted in conferring disease resistance to several types of pathogens and pests. The mutation in NRC4a was previously reported. Importantly, slnrc4a mutant described in this manuscript shows enhanced defense responses such as transcriptional and metabolic changes.

The enhanced disease resistance of slnrc4a mutant to multiple pathogens is interesting observation. However, this manuscript lacks mechanistic investigation. In particular, there is no clear demonstration on why the short variant of NRC4a confers enhanced defense responses and pathogen resistance. In slnrc4a mutant, what is the mRNA and protein levels of NRC4a mutant variant? Has the mutation caused an increased mRNA or protein stability? Does the short variant interferes or activates with other NRC or NLR variant function? It was shown by the same group that SINRC4, in particular CC domain, physically associates with PRR. Does the short form (a part of CC domain) of SINRC4 associate with LeEIX2 or FLS2? Unfortunately, without designing of experiments to elucidate the mechanistic basis of enhanced resistance conferred by SINRC4a truncated variant, the current version of manuscript seems to be descriptive and speculative."

Many thanks for this comment. We wish to point out that the mRNA levels of NRC4a, and its closest homolog NRC4b in our gain of function mutant were previously provided in Leibman-Markus et. al., 2018, in Supplemental Figure 6. There, we observed that the **mRNA levels of NRC4a and NRC4b in the mutant are similar to those observed in the WT M82 background**, indicating that mRNA stability is not significantly compromised or enhanced (although we cannot completely rule out the possibility that it could be altered, with feedback mechanisms preserving **overall** mRNA levels). A comment to this affect was added in the text (lines 254-255). Figure S6 from Leibman-Markus et. al., 2018, is added here for convenience.

Figure S6.

a. Chromatogram of the insertion present in the *slnrc4a*- TL2 and TL5 CRISPR plants. The red bar shows the single nucleotide insertion leading to a 67 aa mutated and truncated protein for SINRC4a.

b. Effect of *SINRC4a* CRISPR-Cas9 editing on *SINRC4a* and *SINRC4b* transcripts levels. RT-PCR analysis confirming no changes in SINRC4a and SINRC4b transcripts levels. Genes were normalized to *SICyclophilin* (*Solyc01g111170*) in the M82 background as a reference gene. Error bars represent the average \pm SEM values of 3 independent experiments, $n = 9$ each. No significant differences with the control were found.

Unfortunately, determining that amount of the native truncated protein is not currently possible. In transient 35S driven expression assays of the 67 a.a. peptide, which was sufficient to exert an increase in defense responses (Leibman-Markus et al., 2018), expression of the 67 a.a. peptide is similar to the Free-mCherry control, while the full length SINRC4a protein is expressed at lower levels. We added Supplemental Figure S5, in which panel (b) addresses this. See below.

We are unable to determine whether the 67 a.a. peptide binds significantly to the LeEIX2 PRR, as the peptide is very highly expressed in transient conditions when compared to the native protein. Calibration experiments were unsuccessful, and we were not able to achieve similar expression levels of the 67 a.a. peptide and the PRR to carry out a Co-IP experiment. We cannot definitively state that specific binding

between the 67 a.a. peptide and the LeEIX2 PRR occurs, although, as the reviewer mentioned, we did show that the coiled-coil domain, which contains the short peptide with an additional ~60 amino acids, does specifically bind to LeEIX2 (Leibman-Markus et. al., 2018).

To further address a possible mechanism, we added Supplemental Figure S5:

Figure S5: The 67 aa *slnrc4a* peptide affects LeEIX2 in a ligand independent manner

N. benthamiana leaves transiently expressing LeEIX2-GFP and free mCherry (Control), the full SINRC4a-mCherry, or the predicted 67 amino acid-mCherry (peptide present in the *slnrc4a* mutant) as indicated, were treated with EIX ($1 \mu\text{g g}^{-1}$ tissue) or water (mock) at the petiole 40 hours after transformation.

(a) LeEIX2-GFP endosomes were visualized by confocal microscopy 15 minutes post EIX treatment. LeEIX2-GFP endosome density with (+) and without (-) EIX was quantified using 3D object counter (Fiji-ImageJ). Error bars represent the average \pm SEM of four independent replicates, five images each. Letters indicate significant differences from the control, two-tailed t-test.

(b) mCherry, SINRC4a-mCherry, or the predicted 67 amino acid-mCherry protein expression level (mean pixel intensity of mCherry signal) was quantified using FIJI-ImageJ. Sixteen images from four experiments were analyzed. Error bars represent the average \pm SEM. Asterisk indicates significant difference (two-tailed t-test, $P < 0.05$).

Overexpression of the full length NRC4 increases LeEIX2 presence on endosomes under basal conditions (Supp. Fig. S5a), and the cells respond further to EIX with an increase in LeEIX2 endosomal presence. Interestingly, overexpression of the 67 aa peptide increases LeEIX2 presence on endosomes similar to the full length peptide under basal conditions, however, in accordance with the MAMP-independent responses we observed in the *slnrc4a* mutant, the cellular response to EIX is diminished. Protein expression of the 67 a.a. peptide under transient conditions is similar to the Free-mCherry control, while the full length SINRC4a protein is expressed at lower levels (Supp. Fig. S5b). Interestingly, treatment with the MAMP EIX increased the level of the Full NRC4 protein but not of the 67 aa peptide (Supp. Fig. S5b), as we previously demonstrated (Leibman-Markus et al., 2018). This result

corresponds with those obtained under basal conditions in the *slnrc4a* mutant, suggesting that the 67 aa peptide does not require EIX to mediate some, though not all, of its effects.

Our data indicates that while the SINRC4a 67 amino acid peptide can respond to the MAMP EIX in some cases (Fig. 6, S2, S3), it does not require MAMPs in order to promote an increase in certain aspects of basal immunity or disease resistance (Fig. 2,7). MAMPs mediate signaling via their PRRs. We previously showed that SINRC4a overexpression can promote an increase in LeEIX2 presence on endosomes (Leibman-Markus et al., 2018), where PRR signaling is taking place (Sharfman et al., 2011). All these data are discussed in the revised manuscript (lines 269-283 and 361-366).

Reviewer #3:

In this interesting study the authors further their investigation of a truncated NRC4a allele. In a previous study (PCE 2018) the authors generated Crispr/cas9 truncation mutants of SINRC4a and found that these plants showed an enhanced defense response to the EIX elicitor and increased resistance to infection of *Botrytis cinerea*. Here, they show that these plants exert heightened resistance to a range of pathogens including bacteria (*Pst* and *Xcv*), bio- and necrotrophic fungi (*Bc* and *Sc*) and two pests (whitefly and tomato leaf miner). Increased resistance correlates with induction of defense markers such as ethylene, defense gene (PR genes and NRCs) expression, terpenoid production and callose depositions, but not ROS production or ion leakage. Upon MAMP treatment or *B. cinerea* infection the *slnrc4* mutants show a potentiated defense response including heightened ROS burst, ethylene production and callose deposition. Surprisingly, the *slnrc4* mutants do not seem to exert pleiotropic phenotypes as their yield in optimal lab/greenhouse conditions are not different than those of wild type plants. Biochemical fractionation of the putative NRC4 fragment produced in the *slnrc4* mutant indicates that it is membrane localized.

Based on the presented data the authors propose that NRC4 mutants might be applicable in agricultural setting to confer broad spectrum resistance to plant pathogens without negatively affecting yield.

The paper is well written and the conclusions are based on a large set of high-quality experiments. Nevertheless, I have some comments for consideration.

Major comments:

1. "The authors claim that the *slnrc4* mutant is a gain-of-function mutation. The proposedly new function that the authors assign to it is that it causes a constitutive priming phenotype explaining the increased resistance to various pathogens and pests.... I do not agree with this concept. The formal definition used in the community for priming is that plants are capable to enhance their basal defense strategies against harmful organisms upon the perception of certain stimuli. This enhanced resistance is not necessarily accompanied by direct activation of defenses, but often depends on a sensitization of the plant tissue to express defenses faster and/or stronger when the plant is under attack. As the *slnrc4* mutants clearly show enhanced basal levels of defence activation in the absence of any pathogen or biotic stress actor this finding disqualifies the phenotype as being primed... Indeed, not the full spectrum of defence outputs is activated... as the plants are sensitized and do respond faster upon pathogen/map perception some sort of priming might also be induced in these plants, but in addition they exert an autoimmune phenotype. I would suggest the authors to adjust their claims in abstract, intro an discussion that states that the mutant induces a priming phenotype."

Many thanks for this comment. We have ourselves debated how to define the observed mutant phenotype, though we still think it is a gain of function phenotype because it is obtained also with transient overexpression of the truncated peptide (Leibman-Markus et. al., 2018). We have amended the title text as suggested, to reflect that the mutant possess increased basal defense as an intrinsic property, regardless of receptor activation, with priming of certain defenses as a possibility

(e.g., lines 1, 23-25, 86-87, 112, 148-149, 177-178, 190-191, 193-194, 210, 256, 267, etc.).

2. "Two *slnrc4* mutants have been generated that differ in a snp in the coding sequence resulting in the induction of a stop codon. In the manuscript it is not clear whether the mutant show the exact same phenotypes and which mutant is used in which assay... It would strengthen the study if the authors can show both mutants in all panels as if they give the same output in all cases the phenotype is likely to be caused by the mutation in *nrc4* and not by another linked mutation in the genome... Are both mutants derived from an independent transformation or do they originate from the same callus/line and hence are not truly independent."

Each mutant is derived from an independent transformation event. Both are predicted to generate a 67 amino acid peptide with one amino acid difference among them, stemming from a different 1 base insertion, but ultimately resulting in the generation of a stop codon at the same position (see also Leibman-Markus et. al., 2018). Since both mutant lines display mostly identical behavior, we use them both randomly, with most of the results included in the manuscript generated using a random mix of both lines. We have added new Supplemental Figure 1, detailing the results obtained with each line separately in a variety of experiments included in the paper (viz., disease resistance, basal immunity in steady state, and agronomic traits).

Figure S1: Enhanced disease resistance, enhanced basal defense parameters, and agronomic traits of *slnrc4a* line#2 and *slnrc4a* line#5.

Select parameters from Figure 1- pathogen resistance (a-d), Figure 2- steady state defense parameters (e-g), and Figure 5- agricultural traits (h-n), graphed separately for two independent *slnrc4a* mutant lines: *slnrc4a-2* (black bars) and *slnrc4a-5* (gray bars).

(a-b) Lesion area was measured 3 days after inoculation with *B. cinerea* (10×10^6 spores/mL), **(a)**, or *S. sclerotium* **(b)**. *O. neolycopersici* infection was measured as percentage of infected leaf out of total leaf area **(c)**. **(d)** Infestation was determined by counting number of insects per leaf and measuring % of infected leaf area two-weeks after *T. absoluta* exposure. **(a-d)** Average \pm SEM of 3-4 independent replicates is shown. Asterisks represent statistical significance in t-test with Welch's correction (*, p-value <0.05; **, p-value <0.01; ***, p-value <0.001).

(e-g) Ethylene production of M82 and *slnrc4a* samples was measured using gas-chromatography **(e)**. M82 average ethylene production is defined as 100%. Average \pm SEM of 5 independent experiments is presented. Letters represent statistical significance in t-test with Welch's correction. **(f)** Conductivity levels of M82 and *slnrc4a* samples immersed in water for 24 h was measured. Average \pm SEM of 4 independent replicates is shown (one-way ANOVA, no significant difference) **(g)** Gene expression analysis of pathogen responsive genes in M82 and *slnrc4a* plants was measured by RT-qPCR. Relative expression normalized to M82. Average \pm SEM of three independent replicates is shown. Asterisks represent statistical significance in t-test with Welch's correction comparing each gene (*, p-value <0.05; **, p-value <0.01; ***, p-value <0.001).

3. "L133-149 The authors monitor expression of a number of defence marker genes such as PR proteins that are proxies for JA, SA and or ISR/SAR. It would strengthen the study if also expression of the PRR FLS2 would be monitored as defence induction typically also correlates with induced accumulation and concomitant expression of PRRs. Information about PRR FLS2 expression (together with that of EKS) would provide insight whether the observed resistance is due to a mild autoimmune phenotype and heightened PRR expression or try priming, in which these receptors are not induced."

We have accepted the reviewers' position on priming. Other than RLK-EKS, we do not currently have a robust set of three biological repeats with additional PRRs. However, we would like to point out that the auto-activation phenotype in *slnrc4a* was preserved under sterile conditions (Figure 7d-f), suggesting that it is not dependent on PRR activation. This was one of the reasons why we categorized this phenotype as intrinsically primed.

4. "L166-170, an increased size of glandular trichomes is observed. This aspect is not discussed further in the manuscript. It this a phenotype more often seen in autoimmune phenotypes and or during priming. Please elaborate on this unexpected finding in the discussion of the manuscript."

Thanks For this comment. Type VI trichomes are the most abundant trichome type on leaves and stems of tomato plants, and significantly contribute to plant defense. The volatile compounds and proteinases produced in these trichomes have been reported in many studies to be toxic to insects, fungi, and bacteria. Many works have reported the mechanism underlying the defensive functions of trichomes in the Solanaceae as Jasmonic acid (JA) dependent. Upregulation or treatment with JA can increase the abundance of trichomes, which we did not observe in *slnrc4a*, or the

amount of metabolites produced, which we did observe in *snrc4a*, as well as activating plant defensive pathways, leading to increased pest and pathogen resistance. However, though several reports indicate that the amount of defensive metabolites produced is a function of the trichome head size, as we report here, few works have related directly to the connection between trichome head size and pathogen resistance or plant defense pathways. Although most reports concerning the connection between trichomes and defense point to JA dependent mechanisms, JA and SA activated pathways are not necessarily discrete in tomato, therefore, attributing changes in metabolite production to ISR or SAR is difficult. We have added some discussion in the manuscript, in both the results and discussion sections, to this effect (see lines 166-169, 295-308).

5. "L203, please show the data to support the claim that plants showing more severe disease symptoms show higher expression of the defence marker genes".

This comment relates specifically to the genes Chitinase (corrected from PR1a) and PI2, as indicated in the text (lines 206-208). The reference is included (Meller-Harel 2014 et. al.), and an additional reference was added (Mehari et al., 2015). In the first work, the biocontrol agent *T. harzianum* was given to tomato plants prior to infection with *B. cinerea*, reducing disease symptoms. The expression of the genes Chitinase and PI2 was reduced in plants pre-treated with *T. harzianum* as compared with those infected with *B. cinerea* without *T. harzianum* pre-treatment. As indicated in the text, this is only a correlation. Genes termed "defense genes" are very diverse and their response to pathogen infection varies.

6. "L215-225, the claim that the plants are gmo free does not correspond with current legislation in e.g. Europe as the mutants have been generated using a technology that uses genetic modification. Please correct this paragraph and related section into that *snrc4* plants that do not contain transgenes retain the phenotype. Consider to mention that these results confirm that the observed phenotype are not due to a mutation due to insertion of the T-DNA. As it is unclear how the mutants have been made this could be an alternative explanation for the observed findings".

Thanks for this comment. The mutants were generated as previously described (Leibman-Markus et. al., 2018), but indeed, retaining similar behavior in the CAS9 free line, which is not considered GMO in some countries, confirms that the phenotype is due to the mutation in NRC4 and not due to a disruption in the genome at the site of the entry of the transgene. The text has been amended as suggested (lines 223-232).

7. "L247 states:" As we demonstrated here that the 67 amino acid CRISPR generated peptide induces a constitutively primed state". Besides that priming is not shown it is also not shown that the 67 aa protein is stably produced and accumulates in the mutants. Please tune down this claim, or show accumulation of the peptide (or at least expression of the mRNA) in the mutant lines."

mRNA levels of NRC4a, and its closest homolog NRC4b, in our gain of function mutant, were previously provided in Leibman-Markus et. al., 2018, in Supplemental Figure 6. There, we observed that the **mRNA levels of NRC4a and NRC4b in the mutant are similar to those observed in the WT M82 background**. A comment to this affect was added in the text. See response to Reviewer #2 above.

8. "L249, "we proceeded to examine whether this peptide retains membranal localization". To me it is unclear where "retains" refers to. What are the assumptions that a membrane localization is foreseen? Please provide reference and explain, or rephrase."

The full length NRC4 protein is partially plasma membrane associated, and can interact there with the PRRs LeEix2 and FLS2- see Leibman-Markus et. al., 2018 (interaction confirmed in the triton-soluble membrane (TSM) fraction). This is referenced in the text. NRC4 was originally identified by us as an interacting partner for LeEIX2 in the TSM fraction of the cell. The text was amended accordingly (lines 253-255).

9. " L251, "showing that it retain the ability to..." is over interpretation of the data. That the protein can be modeled on the ZAR1 structure does not mean that is has the properties and has the same confirmation. Please rephrase/remove."

The text was amended accordingly (lines 257-262).

10. "L254, also in planta grown plants can experience stress and perceive environmental stimuli. Consider rephrasing this part, in e.g. are not exposed to biotic stresses or to MAMPs".

Rephrased accordingly (lines 264-265).

11. L303-311, an equal likely explanation for the observed defense in the ability to of the longer and shorter CC fragment to trigger immune activation is that the C-terminal part of the CC fragment has an autoinhibitory activity. This would not require any additional protein-protein interactions and its removal would potentiate the immune response resulting in pcd.

Amended accordingly (lines 327-328, 347-349). The focus on protein-protein interactions was in the context of the resistosome structure.

"Minor comments:

L 63, citation Ron and Avni incomplete

L124 remove "."

L184-186, Overinterpretation of the data. A correlation between enhanced defense responses and increased resistance is not the same as a causal relation. See also above my remark about priming.

L188, "priming can result in improved disease outcomes". Please adjust, in e.g. in priming can result in improved resistance, or priming can result in decreased disease outcomes.

L246, remove d in and.

Fig S1, include in these graphs also the non EIX treated control for comparison.

L606, include journal."

All minor comments were addressed in the text. Supplemental Figure S2 (previously S1) was amended accordingly.

Harel, Y.M., Mehari, Z.H., Rav-David, D., and Elad, Y. (2014). Systemic Resistance to Gray Mold Induced in Tomato by Benzothiadiazole and *Trichoderma harzianum* T39. *Phytopathology* 104: 150–157.

Leibman-Markus M., Pizarro L., Schuster S., Lin Z.J.D., Gershony O., Bar M., Coaker G., and Avni A., (2018) The intracellular nucleotide binding leucine-rich repeat receptor - SINRC4a enhances immune signaling elicited by extracellular perception. *Plant, Cell & Environment*: 1–15.

Mehari, Z.H., Elad, Y., Rav-David, D., Graber, E.R., and Meller Harel, Y. (2015). Induced systemic resistance in tomato (*Solanum lycopersicum*) against *Botrytis cinerea* by biochar amendment involves jasmonic acid signaling. *Plant and Soil* 395: 31–44.

Sharfman M., Bar M., Ehrlich M., Schuster S., Melech-Bonfil S., Ezer R., Sessa G., and Avni A., (2011) Endosomal signaling of the tomato leucine-rich repeat receptor-like protein LeEix2. *The Plant Journal* 68: 413–423.

Reviewers' comments:

Reviewer #1 (Remarks to the Author):

The authors responded my concerns properly and revised the manuscript based on all the comments from the three reviewers. The quality of the manuscript was improved. I have no further comment regardign to the revised manuscript.

Reviewer #2 (Remarks to the Author):

Authors have improved the quality of manuscript by providing additional data and explanation. Authors responded adequately and I am satisfied with the revised manuscript.

Reviewer #3 (Remarks to the Author):

Resubmission: A gain of function mutation in SINRC4a primes broad spectrum disease resistance.
Pizarro et al
Nat comm

In this resubmission the authors have provided new data and addressed many of the reviewer comments. I am happy to see that most of my comments have been dealt with and together with the other changes made to the manuscript this resulted in a much-improved manuscript. Nevertheless, there are a few issues remaining that have to be dealt with. As mentioned most of my comments have been addressed, but few need additional attention and some new issues emerge as detailed below. Please find my response to the addressed comments marked with a ">" followed by some other comments.

1. "The authors claim that the slnrc4 mutant is a gain-of-function mutation. The proposedly new function that the authors assign to it is that is causes a constitutive priming phenotype explaining the increased resistance to various pathogens and pests.... I do not agree with this concept. The formal definition used in the community for priming is that plants are capable to enhance their basal defense strategies against harmful organisms upon the perception of certain stimuli. This enhanced resistance is not necessarily accompanied by direct activation of defenses, but often depends on a sensitization of the plant tissue to express defenses faster and/or stronger when the plant is under attack. As the slnrc4 mutants clearly show enhanced basal levels of defence activation in the absence of any pathogen or biotic stress actor this finding disqualifies the phenotype as being primed... Indeed, not the full spectrum of defence outputs is activated... as the plants are sensitized and do respond faster upon pathogen/map perception some sort of priming might also be induced in these plants, but in addition they exert and autoimmune phenotype. I would suggest the authors to adjust their claims in abstract, intro an discussion that states that the mutant induces a priming phenotype."

Many thanks for this comment. We have ourselves debated how to define the observed mutant phenotype, though we still think it is a gain of function phenotype because it is obtained also with transient overexpression of the truncated peptide (Leibman-Markus et. al., 2018). We have amended the title text as suggested, to reflect that the mutant possess increased basal defense as an intrinsic property, regardless of receptor activation, with priming of certain defenses as a possibility Pizarro et. al.

(e.g., lines 1, 23-25, 86-87, 112, 148-149, 177-178, 190-191, 193-194, 210, 256, 267, etc.).

– I agree with the concept of the mutation resulting in a gain of function rather than a loss of function mutation, however the concept of priming does not apply. I am happy to see that the authors agree and have adjusted the text in many places. However, the discussion still starts with claiming that the *slnrc4a* mutation resembles priming (L289 and before), while at the end of the chapter it is concluded that the mutation causes an autoimmune phenotype (l307). This is inconsistent, the phenotype does not resemble priming as defence outputs are constitutively activated in the absence of a pathogen as is also the model proposed in fig 7.

2. "Two *slnrc4* mutants have been generated that differ in a snp in the coding sequence resulting in the induction of a stop codon. In the manuscript it is not clear whether the mutant show the exact same phenotypes and which mutant is used in which assay... It would strengthen the study if the authors can show both mutants in all panels as if they give the same output in all cases the phenotype is likely to be caused by the mutation in *nlrc4* and not by another linked mutation in the genome... Are both mutants derived from an independent transformation or do they originate from the same callus/line and hence are not truly independent."

Each mutant is derived from an independent transformation event. Both are predicted to generate a 67 amino acid peptide with one amino acid difference among them, stemming from a different 1 base insertion, but ultimately resulting in the generation of a stop codon at the same position (see also Leibman-Markus et. al., 2018). Since both mutant lines display mostly identical behavior, we use them both randomly, with most of the results included in the manuscript generated using a random mix of both lines. We have added new Supplemental Figure 1, detailing the results obtained with each line separately in a variety of experiments included in the paper (viz., disease resistance, basal immunity in steady state, and agronomic traits).

> thank you for clarifying this, maybe you can also adjust line 318 in that not a mutant has been generated, but that two independent mutants have been generated and were characterised.

3. "L133-149 The authors monitor expression of a number of defence marker genes such as PR proteins that are proxies for JA, SA and or ISR/SAR. It would strengthen the study if also expression of the PRR FLS2 would be monitored as defence induction typically also correlates with induced accumulation and concomitant expression of PRRs. Information about PRR FLS2 expression (together with that of EKS) would provide insight whether the observed resistance is due to a mild autoimmune phenotype and heightened PRR expression or try priming, in which these receptors are not induced."

We have accepted the reviewers' position on priming. Other than RLK-EKS, we do not currently have a robust set of three biological repeats with additional PRRs.

However, we would like to point out that the auto-activation phenotype in *slnrc4a* was preserved under sterile conditions (Figure 7d-f), suggesting that it is not dependent on PRR activation. This was one of the reasons why we categorized this phenotype as intrinsically primed.

> it is pity that these data are not available as in my opinion they would have strengthened the study and could have provided mechanistic insight explaining the enhanced immune response. Increased

PRR expression correlates with enhanced immune outputs as also recently shown in Niko Geldners Cell paper. Would also address a concern of the other reviewer in that the paper is rather descriptive and does not provide mechanistic insight.

4. "L166-170, an increased size of glandular trichomes is observed. This aspect is not discussed further in the manuscript. Is this a phenotype more often seen in autoimmune phenotypes and or during priming. Please elaborate on this unexpected finding in the discussion of the manuscript."

Thanks For this comment. Type VI trichomes are the most abundant trichome type on leaves and stems of tomato plants, and significantly contribute to plant defense. The volatile compounds and proteinases produced in these trichomes have been reported in many studies to be toxic to insects, fungi, and bacteria. Many works have reported the mechanism underlying the defensive functions of trichomes in the Solanaceae as Jasmonic acid (JA) dependent. Upregulation or treatment with JA can increase the abundance of trichomes, which we did not observe in *slnrc4a*, or the amount of metabolites produced, which we did observe in *slnrc4a*, as well as activating plant defensive pathways, leading to increased pest and pathogen resistance. However, though several reports indicate that the amount of defensive metabolites produced is a function of the trichome head size, as we report here, few works have related directly to the connection between trichome head size and pathogen resistance or plant defense pathways. Although most reports concerning the connection between trichomes and defense point to JA dependent mechanisms, JA and SA activated pathways are not necessarily discrete in tomato, therefore, attributing changes in metabolite production to ISR or SAR is difficult. We have added some discussion in the manuscript, in both the results and discussion sections, to this effect (see lines 166-169, 295-308).

> thank you for clarifying. Discussing the correlation between JA signaling and increased metabolite production (L296-308) is interesting and relevant for the study. However, the comment on the relation with ISR/SAR (L308) and metabolites to this study is not clear to me. Maybe this confusion is a consequence of the first part of the discussion not being updated to the concept that the mutation causes a specific autoimmune response rather than priming due to ISR/SAR? Please clarify.

5. "L203, please show the data to support the claim that plants showing more severe disease symptoms show higher expression of the defence marker genes".

This comment relates specifically to the genes Chitinase (corrected from PR1a) and PI2, as indicated in the text (lines 206-208). The reference is included (Meller-Harel 2014 et. al.), and an additional reference was added (Mehari et al., 2015). In the first work, the biocontrol agent *T. harzianum* was given to tomato plants prior to infection with *B. cinerea*, reducing disease symptoms. The expression of the genes Chitinase and PI2 was reduced in plants pre-treated with *T. harzianum* as compared with those infected with *B. cinerea* without *T. harzianum* pre-treatment. As indicated in the text, this is only a correlation. Genes termed "defense genes" are very diverse and their response to pathogen infection varies.

– Thanks for adding the citations.

6. "L215-225, the claim that the plants are gmo free does not correspond with current legislation in e.g. Europe as the mutants have been generated using a technology that uses genetic modification. Please correct this paragraph and related section into that *slnrc4* plants that do not contain transgenes retain the phenotype. Consider to mention that these results confirm that the observed phenotype are not due to a mutation due to insertion of the T-DNA. As it is unclear how the mutants have been made this could be an alternative explanation for the observed findings".

Thanks for this comment. The mutants were generated as previously described (Leibman-Markus et. al., 2018), but indeed, retaining similar behavior in the CAS9 free line, which is not considered GMO in some countries, confirms that the phenotype is due to the mutation in NRC4 and not due to a disruption in the genome at the site of the entry of the transgene. The text has been amended as suggested (lines 223-232).

– Thanks for providing this information, makes the study more complete

7. "L247 states:" As we demonstrated here that the 67 amino acid CRISPR generated peptide induces a constitutively primed state". Besides that priming is not shown it is also not shown that the 67 aa protein is stably produced and accumulates in the mutants. Please tune down this claim, or show accumulation of the peptide (or at least expression of the mRNA) in the mutant lines."

Pizarro et. al.

COMMSBIO-20-0359-T

11

mRNA levels of NRC4a, and its closest homolog NRC4b, in our gain of function mutant, were previously provided in Leibman-Markus et. al., 2018, in Supplemental Figure 6. There, we observed that the mRNA levels of NRC4a and NRC4b in the mutant are similar to those observed in the WT M82 background. A comment to this affect was added in the text. See response to Reviewer #2 above.

> providing this information makes the study more complete.

8. "L249, "we proceeded to examine whether this peptide retains membranal localization". To me it is unclear where "retains" refers to. What are the assumptions that a membrane localization is foreseen? Please provide reference and explain, or rephrase."

The full length NRC4 protein is partially plasma membrane associated, and can interact there with the PRRs LeEix2 and FLS2- see Leibman-Markus et. al., 2018 (interaction confirmed in the triton-soluble membrane (TSM) fraction). This is referenced in the text. NRC4 was originally identified by us as an interacting partner for LeEIX2 in the TSM fraction of the cell. The text was amended accordingly (lines 253-255).

> thank you for providing this information, helps to understand the rationale of the experiment.

9. " L251, "showing that it retain the ability to..." is over interpretation of the data. That the protein can be modeled on the ZAR1 structure does not mean that it has the properties and has the same confirmation. Please rephrase/remove."

The text was amended accordingly (lines 257-262).

> oke

10. "L254, also in planta grown plants can experience stress and perceive environmental stimuli. Consider rephrasing this part, in e.g. are not exposed to biotic stresses or to MAMPs".

Rephrased accordingly (lines 264-265).

> oke

11. L303-311, an equal likely explanation for the observed defense in the ability to of the longer and shorter CC fragment to trigger immune activation is that the C-terminal part of the CC fragment has an autoinhibitory activity. This would not require any additional protein-protein interactions and its removal would potentiate the immune response resulting in pcd.

Amended accordingly (lines 327-328, 347-349). The focus on protein-protein interactions was in the context of the resistosome structure.

→ Oke

→

"Minor comments:

L 63, citation Ron and Avni incomplete

L124 remove "."

L184-186, Overinterpretation of the data. A correlation between enhanced defense responses and increased resistance is not the same as a causal relation. See also above my remark about priming.

L188, "priming can result in improved disease outcomes". Please adjust, in e.g. in priming can result in improved resistance, or priming can result in decreased disease outcomes.

L246, remove d in and.

Fig S1, include in these graphs also the non EIX treated control for comparison.

Pizarro et. al.

COMMSBIO-20-0359-T

12

L606, include journal."

All minor comments were addressed in the text. Supplemental Figure S2 (previously S1) was amended accordingly.

>oke

? Other issues:

- L261. The authors argue that sterile grown plants not exposed to mapms or biotic retained increased disease resistance. This argument is based on increased resistance observed upon infection with Botrytis. However, a bioassay by definition is biotic stress and involves mamps, and it is fundamentally impossible to assess increased resistance without a pathogen. Please rephrase the sentence to better describe this experiment.
- L282 and L356, why would the authors expect that the EIX is required for the phenotype? The mutant shows constitutive defence activation and increase resistance to a variety of pathogens and pests. None of the latter produces EIX as elicitor, hence it would have been very unlikely that EIX is the causal trigger.
- L355, it is speculated that the truncated SLNRC4 fragment might be involved in pore formation. An argument against this proposition it that if it would spontaneously induce pore formation in the pm this would result in ion leakage and increased conductivity has not been observed in this study.

Minor

- L59 Explain H-NLR on first use
- L69 check font
- L 72 replace helper-NLr for H-NLR, now different terms are used for the same class of proteins.
- L142 rephrase this sentence, seems a verb is missing
- L249, split this sentence into two sentences.
- L264, this sentence is confusing, how can a peptide respond to something? Please rephrase.
- L354, consider rephrasing, the sentence is difficult to grasp.
- L356-360. Seems like some words are missing, please rephrase for clarity.

We wish to thank to the reviewers for their time and comments, which have improved the manuscript. Follows a "point by point" response.

Reviewers' comments:

Reviewer #1 (Remarks to the Author):

The authors responded my concerns properly and revised the manuscript based on all the comments from the three reviewers. The quality of the manuscript was improved. I have no further comment regardign to the revised manuscript.

Many thanks.

Reviewer #2 (Remarks to the Author):

Authors have improved the quality of manuscript by providing additional data and explanation. Authors responded adequately and I am satisfied with the revised manuscript.

Many thanks.

Reviewer #3 (Remarks to the Author):

In this resubmission the authors have provided new data and addressed many of the reviewer comments. I am happy to see that most of my comments have been dealt with and together with the other changes made to the manuscript this resulted in a much-improved manuscript. Nevertheless, there are a few issues remaining that have to be dealt with. As mentioned most of my comments have been addressed, but few need additional attention and some new issues emerge as detailed below. Please find my response to the addressed comments marked with a ">" followed by some other comments.

1" .The authors claim that the slnrc4 mutant is a gain-of-function mutation. The proposedly new function that the authors assign to it is that is causes a constitutive priming phenotype explaining the increased resistance to various pathogens and pests.... I do not agree with this concept. The formal definition used in the community for priming is that plants are capable to enhance their basal defense strategies against harmful organisms upon the perception of certain stimuli. This enhanced resistance is not necessarily accompanied by direct activation of defenses, but often depends on a sensitization of the plant tissue to express defenses faster and/or stronger when the plant is under attack. As the slnrc4 mutants clearly show enhanced basal levels of defence activation in the absence of any pathogen or biotic stress actor this finding disqualifies the phenotype as being primed... Indeed, not the full spectrum of defence outputs is activated... as the plants are sensitized and do respond faster upon pathogen/map perception some sort of priming might also be induced in these plants, but in addition they exert and autoimmune phenotype. I would suggest the authors to adjust their claims in abstract, intro an discussion that states that the mutant induces a priming phenotype".

Many thanks for this comment. We have ourselves debated how to define the observed mutant phenotype, though we still think it is a gain of function phenotype because it is obtained also with transient overexpression of the truncated peptide (Leibman-Markus et. al., 2018). We have

amended the title text as suggested, to reflect that the mutant possess increased basal defense as an intrinsic property, regardless of receptor activation, with priming of certain defenses as a possibility)e.g., lines 1, 23-25, 86-87, 112, 148-149, 177-178, 190-191, 193-194, 210, 256, 267, etc.(.

< I agree with the concept of the mutation resulting in a gain of function rather than a loss of function mutation, however the concept of priming does not apply. I am happy to see that the authors agree and have adjusted the text in many places. However, the discussion still starts with claiming that the slnrc4a mutation resembles priming (L289 and before), while at the end of the chapter it is concluded that the mutation causes an autoimmune phenotype (I307). This is inconsistent, the phenotype does not resemble priming as defence outputs are constitutively activated in the absence of a pathogen as is also the model proposed in fig 7.

The first part of the discussion was amended accordingly (lines 286-299).

2" .Two slnrc4 mutants have been generated that differ in a snp in the coding sequence resulting in the induction of a stop codon. In the manuscript it is not clear whether the mutant show the exact same phenotypes and which mutant is used in which assay... It would strengthen the study if the authors can show both mutants in all panels as if they give the same output in all cases the phenotype is likely to be caused by the mutation in nlrc4 and not by another linked mutation in the genome... Are both mutants derived from an independent transformation or do they originate from the same callus/line and hence are not truly independent".

Each mutant is derived from an independent transformation event. Both are predicted to generate a 67 amino acid peptide with one amino acid difference among them, stemming from a different 1 base insertion, but ultimately resulting in the generation of a stop codon at the same position (see also Leibman-Markus et. al., 2018). Since both mutant lines display mostly identical behavior, we use them both randomly, with most of the results included in the manuscript generated using a random mix of both lines. We have added new Supplemental Figure 1, detailing the results obtained with each line separately in a variety of experiments included in the paper (viz., disease resistance, basal immunity in steady state, and agronomic traits.(

<thank you for clarifying this, maybe you can also adjust line 318 in that not a mutant has been generated, but that two independent mutants have been generated and were characterised.

Amended as suggested (lines 329-330).

3" .L133-149 The authors monitor expression of a number of defence marker genes such as PR proteins that are proxies for JA, SA and or ISR/SAR. It would strengthen the study if also expression of the PRR FLS2 would be monitored as defence induction typically also correlates with induced accumulation and concomitant expression of PRRs. Information about PRR FLS2 expression (together with that of EKS) would provide insight whether the observed resistance is due to a mild autoimmune phenotype and heightened PRR expression or try priming, in which these receptors are not induced".

We have accepted the reviewers' position on priming. Other than RLK-EKS, we do not currently have a robust set of three biological repeats with additional PRRs. However, we would like to

point out that the auto-activation phenotype in *slnrc4a* was preserved under sterile conditions (Figure 7d-f), suggesting that it is not dependent on PRR activation. This was one of the reasons why we categorized this phenotype as intrinsically primed.

<it is pity that these data are not available as in my opinion they would have strengthened the study and could have provided mechanistic insight explaining the enhanced immune response. Increased PRR expression correlates with enhanced immune outputs as also recently shown in Niko Geldners Cell paper. Would also address a concern of the other reviewer in that the paper is rather descriptive and does not provide mechanistic insight.

We have carried out qRT-PCR experiments on three additional PRRs and added them to the revised version of Figure 2, as panel (h). We demonstrate that 3 out of the 4 PRRs assayed were upregulated significantly. FLS2, although slightly increased, is not significantly upregulated in *slnrc4a* plants. We have also added discussion to the text (lines 148-151; 300-301; 373-375) to relate to this result.

4" .L166-170, an increased size of glandular trichomes is observed. This aspect is not discussed further in the manuscript. It this a phenotype more often seen in autoimmune phenotypes and or during priming. Please elaborate on this unexpected finding in the discussion of the manuscript".

Thanks For this comment. Type VI trichomes are the most abundant trichome type on leaves and stems of tomato plants, and significantly contribute to plant defense. The volatile compounds and proteinases produced in these trichomes have been reported in many studies to be toxic to insects, fungi, and bacteria. Many works have reported the mechanism underlying the defensive functions of trichomes in the Solanaceae as Jasmonic acid (JA) dependent. Upregulation or treatment with JA can increase the abundance of trichomes, which we did not observe in *slnrc4a*, or the amount of metabolites produced, which we did observe in *slnrc4a*, as well as activating plant defensive pathways, leading to increased pest and pathogen resistance. However, though several reports indicate that the amount of defensive metabolites produced is a function of the trichome head size, as we report here, few works have related directly to the connection between trichome head size and pathogen resistance or plant defense pathways. Although most reports concerning the connection between trichomes and defense point to JA dependent mechanisms, JA and SA activated pathways are not necessarily discrete in tomato, therefore, attributing changes in metabolite production to ISR or SAR is difficult. We have added some discussion in the manuscript, in both the results and discussion sections, to this effect (see lines 166-169, 295-308).

<thank you for clarifying. Discussing the correlation between JA signaling and increased metabolite production (L296-308) is interesting and relevant for the study. However, the comment on the relation with ISR/SAR (I308) and metabolites to this study is not clear to me. Maybe this confusion is a consequence of the first part of the discussion not being updated to the concept that the mutation causes a specific autoimmune response rather than priming due to ISR/SAR? Please clarify.

Amended as suggested. The confusing comment was deleted (lines 313-314).

5" .L203, please show the data to support the claim that plants showing more severe disease symptoms show higher expression of the defence marker genes."

This comment relates specifically to the genes Chitinase (corrected from PR1a) and PI2, as indicated in the text (lines 206-208). The reference is included (Meller-Harel 2014 et. al.), and an additional reference was added (Mehari et al., 2015). In the first work, the biocontrol agent *T. harzianum* was given to tomato plants prior to infection with *B. cinerea*, reducing disease symptoms. The expression of the genes Chitinase and PI2 was reduced in plants pre-treated with *T. harzianum* as compared with those infected with *B. cinerea* without *T. harzianum* pre-treatment. As indicated in the text, this is only a correlation. Genes termed "defense genes" are very diverse and their response to pathogen infection varies.

< Thanks for adding the citations.

6" .L215-225, the claim that the plants are gmo free does not correspond with current legislation in e.g. Europe as the mutants have been generated using a technology that uses genetic modification. Please correct this paragraph and related section into that slnrc4 plants that do not contain transgenes retain the phenotype. Consider to mention that these results confirm that the observed phenotype are not due to a mutation due to insertion of the T-DNA. As it is unclear how the mutants have been made this could be an alternative explanation for the observed findings."

Thanks for this comment. The mutants were generated as previously described (Leibman-Markus et. al., 2018), but indeed, retaining similar behavior in the CAS9 free line, which is not considered GMO in some countries, confirms that the phenotype is due to the mutation in NRC4 and not due to a disruption in the genome at the site of the entry of the transgene. The text has been amended as suggested (lines 223-232).

< Thanks for providing this information, makes the study more complete

7" .L247 states:" As we demonstrated here that the 67 amino acid CRISPR generated peptide induces a constitutively primed state". Besides that priming is not shown it is also not shown that the 67 aa protein is stably produced and accumulates in the mutants. Please tune down this claim, or show accumulation of the peptide (or at least expression of the mRNA) in the mutant lines".

mRNA levels of NRC4a, and its closest homolog NRC4b, in our gain of function mutant, were previously provided in Leibman-Markus et. al., 2018, in Supplemental Figure 6. There, we observed that the mRNA levels of NRC4a and NRC4b in the mutant are similar to those observed in the WT M82 background. A comment to this affect was added in the text. See response to Reviewer #2 above.

<providing this information makes the study more complete.

8" .L249, "we proceeded to examine whether this peptide retains membranal localization." To me it is unclear where "retains" refers to. What are the assumptions that a membrane localization is foreseen? Please provide reference and explain, or rephrase".

The full length NRC4 protein is partially plasma membrane associated, and can interact there with the PRRs LeEix2 and FLS2- see Leibman-Markus et. al., 2018 (interaction confirmed in the triton-soluble membrane (TSM) fraction). This is referenced in the text. NRC4 was originally

identified by us as an interacting partner for LeEIX2 in the TSM fraction of the cell. The text was amended accordingly (lines 253-255).

<thank you for providing this information, helps to understand the rationale of the experiment.

9 " .L251, "showing that it retain the ability to..." is over interpretation of the data. That the protein can be modeled on the ZAR1 structure does not mean that is has the properties and has the same confirmation. Please rephrase/remove".

The text was amended accordingly (lines 257-262).

<oke

10" .L254, also in planta grown plants can experience stress and perceive environmental stimuli. Consider rephrasing this part, in e.g. are not exposed to biotic stresses or to MAMPs."

Rephrased accordingly (lines 264-265).

<oke

11 .L303-311, an equal likely explanation for the observed defense in the ability to of the longer and shorter CC fragment to trigger immune activation is that the C-terminal part of the CC fragment has an autoinhibitory activity. This would not require any additional protein-protein interactions and its removal would potentiate the immune response resulting in pcd.

Amended accordingly (lines 327-328, 347-349). The focus on protein-protein interactions was in the context of the resistosome structure.

< Oke

"Minor comments:

L 63, citation Ron and Avni incomplete

L124 remove".“

L184-186, Overinterpretation of the data. A correlation between enhanced defense responses and increased resistance is not the same as a causal relation. See also above my remark about priming.

L188, "priming can result in improved disease outcomes". Please adjust, in e.g. in priming can result in improved resistance, or priming can result in decreased disease outcomes.

L246, remove d in and.

Fig S1, include in these graphs also the non EIX treated control for comparison.

L606, include journal".

All minor comments were addressed in the text. Supplemental Figure S2 (previously S1) was amended accordingly.

<oke

<Other issues:

-L261. The authors argue that sterile grown plants not exposed to mapms or biotic retained increased disease resistance. This argument is based on increased resistance observed upon infection with Botrytis. However, a bioassay by definition is biotic stress and involves mamps, and it is fundamentally impossible to assess increased resistance without a pathogen. Please rephrase the sentence to better describe this experiment.

Rephrased to clarify we are referring to pre-exposure to a biotic stressor prior to pathogen inoculation. (lines 263-265).

-L282 and L356, why would the authors expect that the EIX is required for the phenotype? The mutant shows constitutive defence activation and increase resistance to a variety of pathogens and pests. None of the latter produces EIX as elicitor, hence it would have been very unlikely that EIX is the causal trigger.

Rephrased to clarify we were referring to all MAMPs/ effectors in general in connection with the 67 amino acid peptide activity, and not specifically to EIX; EIX was merely used as an example. (Lines 269; 369).

-L355, it is speculated that the truncated SLNRC4 fragment might be involved in pore formation. An argument against this proposition it that if it would spontaneously induce pore formation in the pm this would result in ion leakage and increased conductivity has not been observed in this study.

The HR induced by the pore structure of the resistosome is somewhat speculative. Possibly, there exists a "partial" or "altered" structure which promotes some forms of defense without causing auto-HR. We do not feel strongly about this and have deleted this sentence.

Minor

-L59 Explain H-NLR on first use

First use is in line 52, and explained.

-L69 check font

Font size mistake was corrected.

-L 72 replace helper-NLr for H-NLR, now different terms are used for the same class of proteins.

Corrected.

-L142 rephrase this sentence, seems a verb is missing

Corrected.

-L249, split this sentence into two sentences.

Corrected.

-L264, this sentence is confusing, how can a peptide respond to something? Please rephrase.

Corrected.

-L354, consider rephrasing, the sentence is difficult to grasp.

Corrected.

-L356-360. Seems like some words are missing, please rephrase for clarity.

Corrected.

REVIEWERS' COMMENTS:

Reviewer #3 (Remarks to the Author):

To authors have provided interesting new data regarding the (increased) expression of PRR encoding genes in the SINRC4a transgenics. These new data increase the impact of the study as they could provide a mechanistic explanation for the observed broad spectrum resistance to various pathogens and pests.

I am pleased with the revision and the adequate respond of the authors to my concerns. I want to congratulate the authors with their interesting findings providing new insights in the function of NRCs in solanacea.